# Deciphering genetic causes for sex differences in human health through drug metabolism and transporter genes

Yingbo Huang [1], Yuting Shan [1], Weijie Zhang[2], Adam M. Lee[1], Feng Li [1,3], Barbara E. Stranger [4] & R. Stephanie Huang [1] ✉

Sex differences have been widely observed in human health. However, little is known about the underlying mechanism behind these observed sex differences. We hypothesize that sex-differentiated genetic effects are contributors of these phenotypic differences. Focusing on a collection of drug metabolism enzymes and transporters (DMET) genes, we discover sex-differentiated genetic regulatory mechanisms between these genes and human complex traits. Here, we show that sex-differentiated genetic effects were present at genome-level and at DMET gene regions for many human complex traits. These sex-differentiated regulatory mechanisms are reflected in the levels of gene expression and endogenous serum biomarkers. Through Mendelian Randomization analysis, we identify putative sex-differentiated causal effects in each sex separately. Furthermore, we identify and validate sex differential gene expression of a subset of DMET genes in human liver samples. We observe higher protein abundance and enzyme activity of CYP1A2 in male-derived liver microsomes, which leads to higher level of an active metabolite formation of clozapine, a commonly prescribed antipsychotic drug. Taken together, our results demonstrate the presence of sex-differentiated genetic effects on DMET gene regulation, which manifest in various phenotypic traits including disease risks and drug responses.

Sex differences have been frequently observed in human health in the form of differences in disease incidence rates, disease progression, and responses to treatment[1-3]. These differences are at least in part related to genetic differences between males and females[4-6]. Two large-scale consortium studies have investigated sex differences in the genetic basis of complex traits[7] and in genetic regulation of gene expression[8]. Both studies demonstrate that sex-specific or sex-differentiated genetic effects can be masked when a genome-wide association study (GWAS) is performed using a sex-combined model[7]. Consequently, sex-differentiated causal inferences—causal effects arising from sex-differentiated genetic effects—can be missed when

using sex-combined analyses. Although these large-scale studies provide strong support for the presence of sex-differentiated genetic regulation, the broad scope of these reports prevents them from elucidating the precise sex-differentiated molecular mechanisms. Without such information, it is difficult to translate these findings into better patient care and apply them towards personalized medicine.

One of the reasons for late phase clinical trial drug development failure is the large variability in treatment response among individuals. Among patient characteristics, biological sex is a major source of inter-individual variability, wherein males and females may respond differently to the same medications[9]. Drug metabolism enzymes and

[1]Department of Experimental and Clinical Pharmacology, College of Pharmacy, University of Minnesota, Minneapolis, MN, USA. [2]Department of Bioinformatics and Computational Biology, University of Minnesota, Minneapolis, MN, USA. [3]eGenesis, Inc, Cambridge, MA, USA. [4]Department of Pharmacology, Center for Genetic Medicine, Northwestern University Feinberg School of Medicine, Chicago, IL, USA. ✉e-mail: rshuang@umn.edu

transporters (DMET) genes influence the pharmacokinetics, pharmacodynamics, and safety profiles of drugs[10, 11]. Pharmacogenomic studies have revealed that genetic variations in DMET genes are strongly correlated with drug response[12, 13]. However, these studies typically employ sex-combined models and report results in a sex-combined fashion[14], because of the sample size/power limitations, hence underestimating the role of sex as a modifier of the drug response. Moreover, functions of DMET genes are critical in determining the amount of endogenous substrates (e.g., serum biomarkers that are frequently used for disease diagnoses[15, 16]). A comprehensive study of sex differences in the genetic basis of DMET genes and their health impact is currently lacking.

In this study, we hypothesized that single-nucleotide polymorphisms (SNPs) located in DMET gene regions contribute to the observed sex-differentiated genetic effects of human complex traits. Such differences could derive from sex differences in the genetic regulation of gene expression and serum biomarkers; and in metabolism of exogenous substrates (drugs); all of which can subsequently impact human health phenotypes. We tested both sex-differential genetic effects where the effect size of genetic regulation is different between sexes, and sex-specific genetic effects where the association only significant in one sex. Using sex-stratified genome-wide association study (GWAS) summary statistics for 564 traits from the UK Biobank (UKBB)[17], we characterized genome-wide sex-differentiated genetic effects on complex traits, and also focused on DMET gene regions. Through sex-stratified expression quantitative trait loci (eQTL) analysis of DMET genes using the Genotype-Tissue Expression (GTEx) project resources and sex-aware mendelian randomization (MR) test focusing on serum biomarkers, we discovered putative sex-differentiated regulatory mechanisms contributing to sex differences in human health. Furthermore, we identified sex-differentially expressed DMET genes in the human liver and linked them to literature reported evidence on sex differences in drug response. Finally, we highlighted sex differences in *CYP1A2* in metabolizing clozapine and experimentally confirmed the differential expression and activity of this gene in the human liver microsomes.

## Results

### Sex differences in genetic architecture and genetic effects of DMET genes

Sex differences in human phenotypes can be driven by sex-differentiated genetic effects[4, 18]. However, such effects can be masked in GWASs because individuals of both sexes have typically been analyzed together[7]. In this study, we evaluated the sex differences in the genetic architecture of human complex traits in each sex separately using sex-stratified GWAS summary statistics from the UKBB (http://www.nealelab.is/uk-biobank, Fig. 1a). We analyzed 564 traits (421 binary/categorical traits, 143 continuous traits, Supplementary Data 2), for which at least one DMET region SNP shows significant trait association based on the GWAS catalog. DMET genes, which encode 222 metabolism enzymes and 150 transporters, were retrieved from a previous publication (Supplementary Data 1)[10]. We first estimated the narrow-sense heritability of each trait separately in each sex to quantify the proportion of phenotypic variance explained by the common genetic variation[19]. Sex differences in trait heritability suggest a different molecular mechanism and/or a different degree of environmental effect between sexes on the trait. Of 564 traits, 83 traits (14.72%) showed significant sex differences in heritability (FDR < 0.05, Fig. 1b and Supplementary Data 3). Among them, 56 traits (67.4%) had higher heritability in males, including neuropsychiatric traits, such as ever attempted suicide and anxiety, and other diseases, such as gout, cardiovascular complications (coronary atherosclerosis) and diabetes. In comparison, traits such as hypothyroidism, osteoporosis and gallbladder disorders have higher heritability in females. These findings are robust across different methods employed to estimate heritability (Supplementary Fig. 1). When expanding the genome-

wide heritability analysis to additional 1222 traits that have sufficient samples for both sexes in UKBB and are not known to related to DMET genetic regions, we found 13.7% (167/1222) of them showing sex differences in global heritability (Supplementary Fig. 2, Supplementary Data 4). Interestingly, similar 13.40% (71/530) traits showing significant differences in their heritability between two sexes have been reported by an independent study[7].

We also estimated genetic correlation between sexes for each trait to evaluate potential sex differences in genetic architecture. Genetic correlation estimates the proportion of variance that two groups share due to genetic causes[20]. Low genetic correlation between males and females in the same trait suggests sex differences in the genetic architecture. We focused on only those heritable traits (defined by greater than median estimated heritability across all traits through sex-combined GWAS analysis) to generate robust estimates of the male-female genetic correlation. We identified 253 traits for which male-female genetic correlation differed significantly from 1 (Supplementary Fig. 3, Supplementary Data 3, FDR < 0.05), suggesting a global difference in genetic architecture between the sexes. These traits with sex-differentiated architecture include gout ($r_g$ = 0.459; FDR = 1.26 × $10^{-9}$), heart attack ($r_g$ = 0.573; FDR = 6.39 × $10^{-10}$), and cholelithiasis ($r_g$ = 0.634; FDR = 1.18 × $10^{-5}$).

For variants located in DMET gene regions, to quantify sex differences in SNP-trait associations, we calculated the z-score and *P* values for each SNP that mapped into the *cis*-DMET regions to each trait. We defined SNPs with sex-differentiated effects (SDEs) as those trait-associated SNPs with a significant sex difference in genetic effects (based on z-score FDR < 0.05). In total, we identified 25 traits harboring at least one SDEs (Fig. 1c, Table 1 and Supplementary Data 5) and mapped 954 SDEs onto 109 tagging loci ($r^2$ < 0.2) and 125 genes. The two disease traits with the largest numbers of SDEs are self-reported gout and hypothyroidism/myxedema (Table 1). We indeed found SDEs mapped in genes previously implicated in disease, such as *ABCG2* in gout[15].

15 traits were found to have sex differences both in heritability (global) and SDEs in DMET gene regions (Fig. 1d), affirming the presence of different genetic effects between males and females. However, whether these differential genetic effects are functionally related to human health is unknown. To test this, we employed colocalization analysis, which assess the probability of two GWAS traits sharing a common causal variant in a genomic region[21]. In another word, if for the same trait, the sex-stratified GWAS results were not colocalized, the probability is high that males and females have a different causal mechanism for that trait. Among the aforementioned 15 traits, we excluded three of them (Treatment code: levothyroxine sodium, treatment code: thyroxine product, major coronary heart disease event excludes revascularization) as they represented information redundant to other traits in the list. All 12 traits exhibit genetic correlation coefficients differing from 1 (Supplementary Fig. 4). We detected 22 male-specific causal loci in traits such as gout and major coronary heart disease events, and 13 female-specific causal loci in traits such as hypothyroidism (Fig. 1e, Supplementary Data 6) after colocalization analysis. All support the presence of sex-differentiated genetic effects of traits in *cis*-DMET gene regions.

Gout and hypothyroidism were found to have higher number of sex-differentiated causal SNPs in the DMET gene regions. Previous findings indicate that accumulation of crystal form of urate at the joint causes gout[22]. We identified male-specific SNP-trait associations of gout in the *SLC22A12* region (Fig. 1f), specifically an upstream variant rs2360872. *SLC22A12*, also known as urate transporter 1, is involved in regulating urate levels in the blood and its variants are associated with serum uric acid level and gout development[23]. Our results suggest that rs2360872 affects gout uniquely in males. Additionally, we detected female-specific SNP-trait associations of hypothyroidism at the 5'UTR of *SLC66A1* (Fig. 1f). *SLC66A1* is a

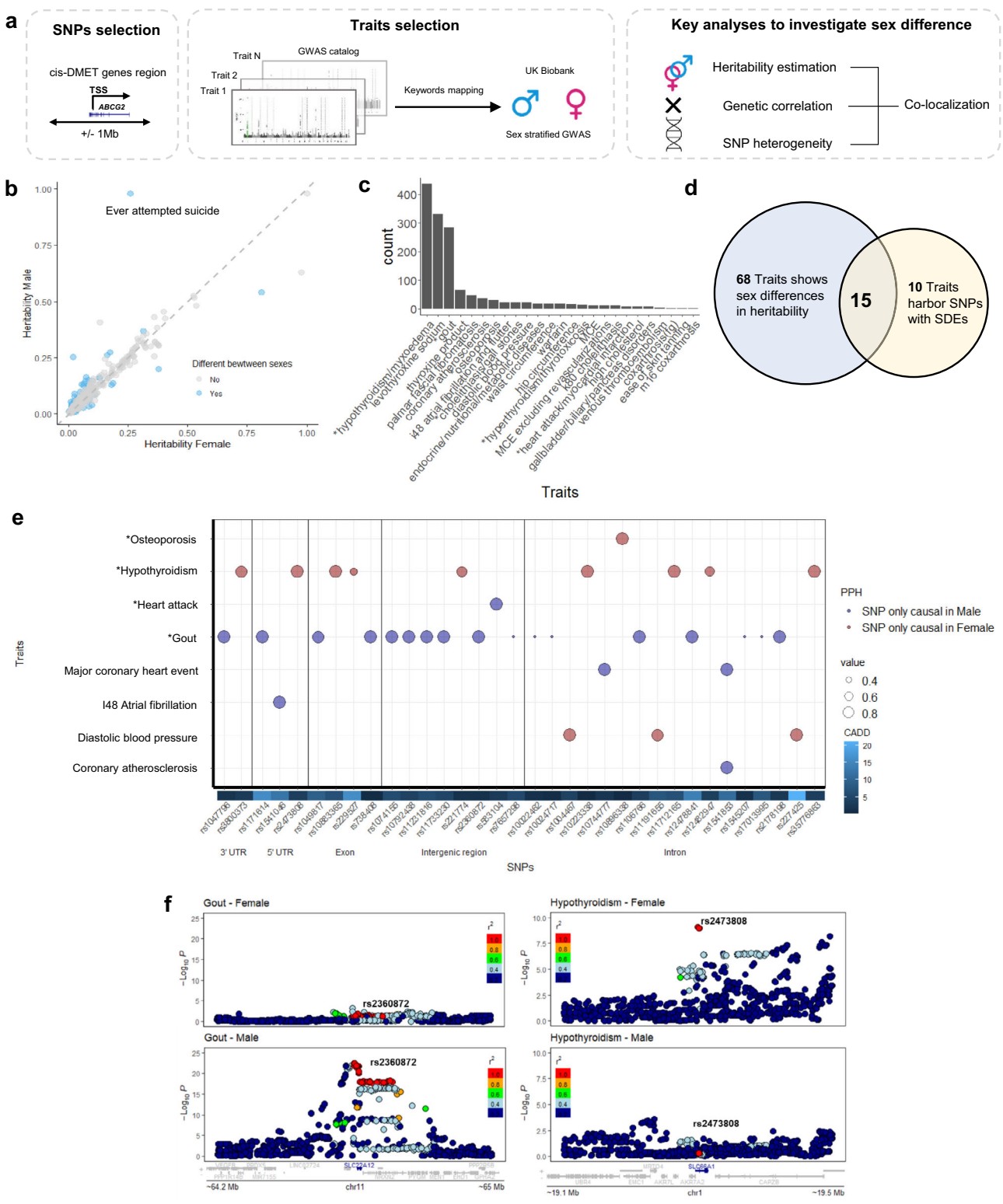

lysosomal amino acid transporter that mediates the export of cationic amino acids from lysosomes[24]. The relationship between *SLC66A1* and thyroid function remains unknown.

## Sex differences in the genetic regulation of gene expression may lead to observed sex differences in human complex traits

Sex-differentiated genetic regulation of gene expression can lead to different manifestations in downstream biological pathways. Therefore, we characterized sex differences of genetic regulation on gene expression in human liver samples using sex-stratified *cis*-eQTLs produced by the GTEx Consortium[25].

Sex-differentiated eQTLs were defined as expression associated variants who have significantly different effect size between males and females (quantified through a z-score testing). We identified 31 sex-differentiated eQTLs (FDR < 0.1) in and near 2 DMET genes (Supplementary Fig. 5, Supplementary Data 7). Among them, rs34109652, is associated with *UGT2B17* gene expression only in males ($\beta_{male} = 0.84$; $P_{male} = 5.44 \times 10^{-7}$; $\beta_{female} = 0.27$; $P_{female} = 0.078$;

**Fig. 1 | Sex-differentiated genetic effect exists in genome-wide and in near-DMET gene regions for complex traits. a** Overview of trait selection and analytical pipeline. We defined each *cis*-DMET gene region as ±1 Mb from the DMET gene transcription start site. We first selected available human complex traits for which sex-stratified GWAS summary statistics are available[17]. The trait list was further narrowed down by the presence of significant SNP-trait associated with the DMET gene regions. In total, we evaluated 564 traits (421 categorical/binary traits and 143 non-binary/non-categorical traits). We then performed several sex-aware analyses (sex-stratified heritability, male-female genetic correlation, sex-differentiated genetic effects) to characterize sex differences both a genome-wide and in DMET gene regions. Lastly, we performed sex-stratified colocalization analysis between sexes to identify DMET variants' unique SNP-traits association in each sex for the traits that demonstrated sex differences in genetic basis. **b** Male and female heritability estimates for 564 traits. Each point represents the estimated heritability for a given trait. Blue indicates the 83 traits with a significant (FDR < 0.05) sex difference in heritability. **c** Number of sex-differentiated effects (SDEs) mapping to DMET genes regions. **d** 15 overlapped traits with SDEs in *cis*-DMET gene regions and a significant sex difference in heritability. **e** Sex-stratified colocalization of GWAS signal for 8 traits. SDEs are labeled on the x-axis, and the traits are labeled on the y-axis. The posterior probability of hypothesis (PPH) 1 (blue, representing SNP putatively causal only in Male) and 2 (red, representing SNP putatively causal only in Female) are color-labeled and the value of PPH is represented by the size of the circle; only PPH > 0.25 is shown. The variant effect and Combined Annotation Dependent Depletion (CADD) score are shown for each SNP. The asterisk indicates self-reported traits. **f** LocusZoom plots of *SLC22A12* (left) and *SLC66A1* (right) from sex-stratified GWAS for self-reported gout and self-reported hypothyroidism, respectively. Linkage disequilibrium (LD) between variants is quantified by the squared Pearson coefficient of correlation ($r^2$).

## Table 1 | Summary of SDEs found in different traits

| Traits | Cis-DMET genes | Number of SDEs (FDR < 0.05) in corresponding cis-DMET gene regions |
|---|---|---|
| Self-reported: hypothyroidism/myxoedema | ALDH2 | 218 |
| Self-reported: gout | ABCG2 | 128 |
| Self-reported: hypothyroidism/myxoedema | SLC16A1 | 108 |
| Self-reported: gout | SLC22A11 | 54 |
| Self-reported: gout | SLC5A6 | 53 |
| Self-reported: hypothyroidism/myxoedema | ALDH8A1 | 48 |
| Self-reported: osteoporosis | ALDH3B1 | 31 |
| Palmar fascial fibromatosis | SLCO5A1 | 29 |
| Self-reported: hypothyroidism/myxoedema | CYP20A1 | 27 |
| i48 atrial fibrillation and flutter | CYP17A1 | 24 |
| Diastolic blood pressure | CYP17A1 | 21 |
| Self-reported: gout | ABCB9 | 20 |
| Endocrine, nutritional and metabolic diseases | ALDH5A1 | 18 |
| Palmar fascial fibromatosis | PPARA | 18 |
| Self-reported: cholelithiasis/gallstones | ABCG5 | 15 |
| Treatment code: warfarin | CYP17A1 | 15 |
| Coronary atherosclerosis | SLC22A2 | 15 |
| Self-reported: gout | SLC29A2 | 15 |
| Coronary atherosclerosis | CYP20A1 | 12 |
| Self-reported: hypothyroidism/myxoedema | ABCC2 | 11 |
| k80 cholelithiasis | ABCG5 | 11 |
| Self-reported: hyperthyroidism/thyrotoxicosis | CYP20A1 | 11 |
| Hip circumference | CYP7B1 | 10 |

Traits with at least 10 SDEs in the cis-DMET gene regions are shown. Traits are sorted by the number of SDEs. Treatment code 'levothyroxine sodium/ thyroxine product' was excluded from the table due to information redundant with self-reported: hypothyroidism/myxedema.

Z-score = −2.95; Fig. 2a). This gene encodes an enzyme that transforms steroid hormones such as testosterone[26]. A previous study reported higher UGT2B17 enzyme activity in males[27]. This male-specific genetic regulation could in part explain the observation of higher *UGT2B17* activity in male, though functional work is required for validation.

In addition to sex-differentiated eQTLs, we also identified 40 potential sex-specific eQTLs in the DMET gene regions (FDR < 0.1 in one sex and FDR > 0.1 in the other sex of eQTL, Fig. 2b, Supplementary Data 7). For example, variant rs854572 on chromosome 7, is associated with *PON1* gene expression in males only ($P_{male} = 4.13 \times 10^{-8}$; $P_{female} = 0.01$; Z-score = 0.33 Fig. 2a). This gene encodes serum paraoxonase and arylesterase 1 enzyme, an enzyme exerting protective effects in Major Adverse Cardiovascular Events (MACE = death, myocardial infarction, stroke). In a sex-combined GWAS, this variant is associated with the activity of paraoxonase and arylesterase 1[28]. Moreover, higher arylesterase

activity has been reported in females[29]. Although sex-specific association of this variant with serum enzyme activity has not been tested, this sex-specific eQTL provides a putative explanation for the observed higher *PON1* serum activity in females. Together, our results support the presence of sex-differentiated genetic regulation of DMET gene expression.

To examine the potential biological consequences of these sex-differentiated/specific eQTLs, we performed colocalization[21] between them and the 564 complex traits described above. We identified 11 traits with a sex-differentiated/specific eQTL colocalization (PPH4 > 0.5 in only a single sex, Fig. 2c, Supplementary Data 8 Binary Traits, Supplementary Data 9 continuous traits). Among them, multiple traits related to alcohol intake are colocalized with a male-specific eQTL for *ADH1C* ($PPH4_{male} = 0.73$, $PPH4_{female} = 0.073$, Fig. 2d). This gene encodes enzymes that metabolize a wide variety of substrates, including alcohol[30]. Alcohol metabolism determines blood alcohol level over time, and the extent of organ exposure to alcohol.

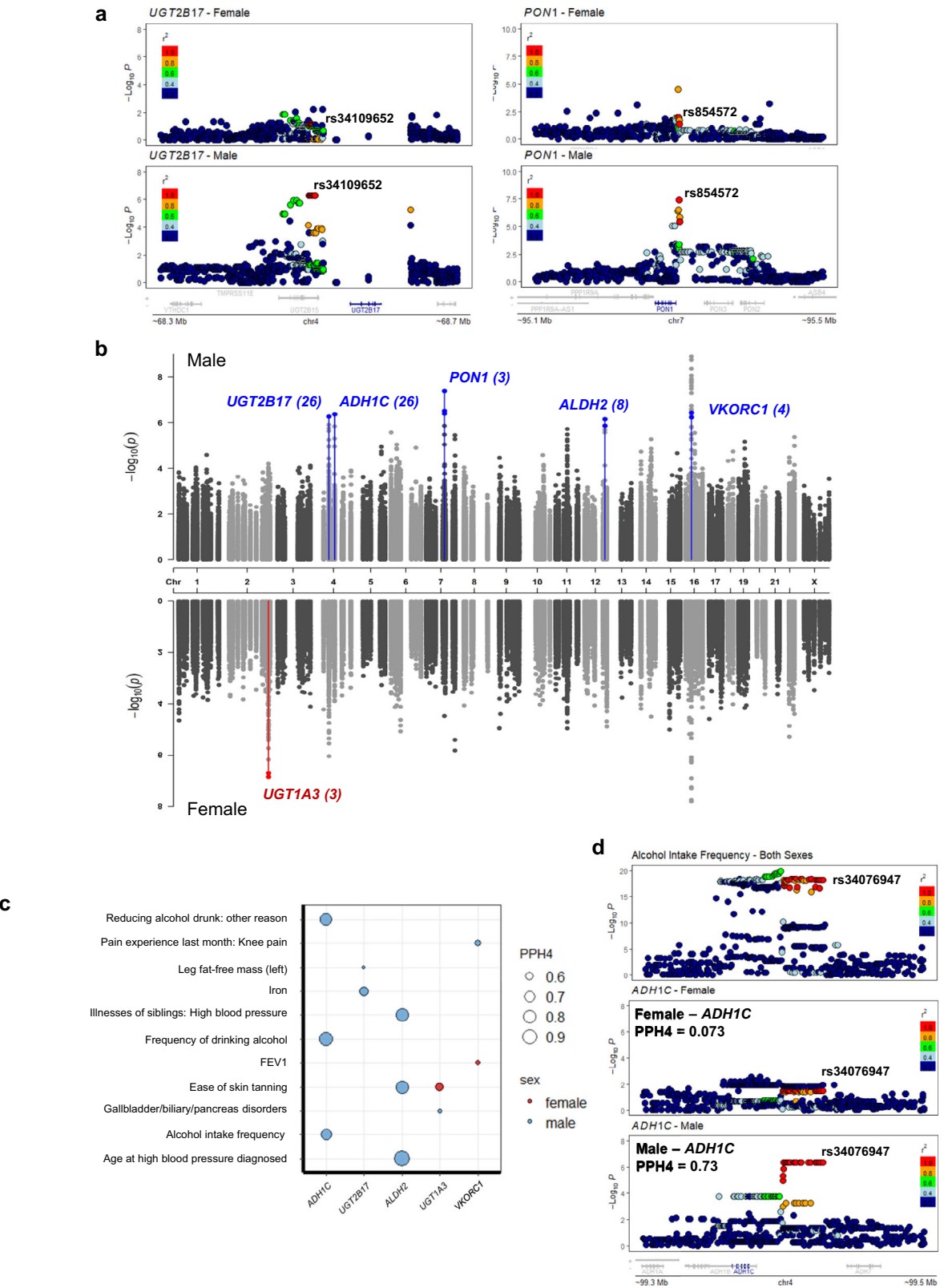

Therefore, *ADH1C* has been implicated in alcohol dependency. Sex differences in alcohol consumption have been widely reported, where males are more likely to drink alcohol and to consume more than females[31]. Genetic variants in *ADH1C* are associated with alcohol metabolism capacity[32]. In males, genetic variants of *ADH1C* showed significant association and a larger effective size with heavy/excessive alcohol drinking habits than females[33]. Our results provided a plausible explanation for this phenomenon where sex-differentiated genetic regulation of *ADH1C* may be the culprit. Importantly, all traits that showed significant sex-dependent colocalization with *ADH1C* eQTLs were related to alcohol drinking frequency, but not alcoholism. This is concordant with previous reports that *ADH1C* genetic variants were not associated with alcoholism[33]. Of note, the observed sex dimorphism in alcohol consumption can be affected by factors, such as body

**Fig. 2 | Sex differences in the genetic regulation of gene expression in human liver in part mediate human complex traits. a** Example of sex-differentiated/specific *cis*-DMET eQTLs in both sexes for *UGT2B17* (left panels) and *PON1* (right panels) in liver. **b** Manhattan plot of sex-stratified cis-eQTLs in DMET regions in human liver. Sex-differentiated/specific *cis*-eQTLs and their corresponding genes are labeled (blue: Male, red: Female) and parentheses indicates the number of *cis*-eQTLs. Highlighted genes are statistically significant after multiple testing correction (FDR < 0.1). **c** Colocalization of GWAS traits with sex-differentiated/specific *cis*-eQTLs for DMET genes. PPH4 values are represented by the size of circles. Only colocalizations with PPH4 > 0.5 in one sex are shown (females: red; males: blue). **d** LocusZoom plots for alcohol intake frequency at the *ADH1C* locus. The top panels illustrate the results from the sex-combined GWAS, while female-only and male-only *cis*-eQTLs are shown in the middle and the bottom panels. rs34076947 is a male-specific *ADH1C cis*-eQTL in liver. In (**a**) and (**d**), linkage disequilibrium between loci is quantified by the squared Pearson coefficient of correlation ($r^2$). In (**a**, **b**, **d**), association testing was performed using a linear regression model by FastQTL.

size, sociocultural behavior. The identification of genetic contributor to this trait should not be interpreted without these other factors. Our results provided a plausible explanation for this phenomenon where sex-differentiated genetic regulation of *ADH1C* may play a part in this complex issue. Overall, our colocalization results linked the genetic basis of a small number of complex traits to sex-dependent genetic effects on expression levels of specific DMET genes.

## Sex differences in the DMET region genetic regulation of human serum biomarkers and their impact on human health outcomes

Clinical laboratory tests are frequently used to diagnose diseases and monitor human health status. Many of these serum biomarkers are transformed in the liver by DMET enzymes[10]. Importantly, sex differences have been known for the level of many serum biomarkers[34]. While the genetic basis of serum biomarkers has been studied in large cohorts in a sex-combined fashion[35], sex differences in genetic basis of those biomarkers and putative causal relationships with human diseases have not been extensively studied. Given the relevance of serum biomarkers in human health and the potential masking effect introduced by the sex-combined model, we hypothesized that sex-differentiated causal relationships exist, and can be revealed by sex-aware characterization of the genetic regulation of serum biomarkers and human disease risks. To test this hypothesis, we performed Mendelian Randomization (MR) in both sex-combined and sex-stratified models. We pre-selected traits that harbor at least 1 significant SNP ($P < 5 \times 10^{-8}$) that are mapped to a DMET gene region. In total, we have 29 serum biomarker traits as exposures and 186 outcomes selected from the 564 human complex traits described above. A sex-specific likelihood of causal relationship is defined if it is significant in only a single sex but not in the sex-combined population ($P < 0.05/(186*29)$ threshold, Fig. 3a).

We first examined the proportion of significant serum biomarker level-associated SNPs ($P < 5 \times 10^{-8}$) that are located in the DMET regions relative to the whole genome (Fig. 3b, Supplementary Data 10); and found the DMET region variants were associated with levels of 29 serum biomarkers. The DMET region serum biomarker associations account for 20% (in the case of c-reactive protein) to 100% (in case of oestradiol) of overall genomic association with these serum biomarkers. The proportion of serum biomarker-associated variants that mapped to *cis*-DMET gene regions is greater than random selection of variants from genome sequence of the same length (Supplementary Fig. 6), which suggests the important impact of DMET genes on serum biomarker levels.

Using the same analytical pipeline described earlier on human complex traits (not including any of the serum biomarker traits), we characterized sex differences in the genetic basis of serum biomarker traits. As expected, we observed that hormone-related traits, such as testosterone, harbor the largest sex differences in estimated heritability (Supplementary Data 11), genetic correlations (Supplementary Fig. 8, Supplementary Data 11), and SDEs (Supplementary Fig. 9, Supplementary Data 12)[34].

By conducting MR analysis using sex-combined and sex-stratified models, we identified a total of 141 female-specific and 167 male-specific putative causal relationships (Fig. 3c, Supplementary Data 13), which were not seen in the sex-combined model. Serum biomarkers,

such as bilirubin and aspartate aminotransferase, have a large number of female-significant causal relationships, whereas testosterone, cholesterol and glucose have a large number of male significant causal relationships (Supplementary Fig. 10). We found sex-specific causal relationships between blood cell counts (e.g., monocyte count, platelet count) and several anthropometric traits (e.g., BMI, waist-to-hip ratio, hip circumference). While sex differences have been known for either the blood cell counts or the anthropometric traits[36, 37], these putative sex-specific causal relationships have not been reported. Among diseases, we identified 24 traits that exhibit at least 1 sex-specific causal relationship (Fig. 3d). We observed that testosterone increases the likelihood of high blood pressure only in females; and apolipoprotein B increases the likelihood of coronary atherosclerosis and major coronary heart disease events only in males. Given that DMET genes are largely involved in the transformation of these endogenous compounds, we hypothesized that genetic variations in DMET gene regions affect levels of these serum biomarkers and result in sex-specific human health outcomes. We further conducted MR analysis only considering SNPs in DMET gene regions in traits with sex-specific causal effect; and identified 61/141 (43.2%) female-specific and 66/167 (39.5%) male-specific putative causal relationships that remained valid (Bonferroni adjusted $P < 0.05$, Supplementary Fig. 11, Supplementary Data 14). Overall, our results highlight the importance of examination of causal relationships in each sex separately, by which sex-specific causal effect can be identified.

To understand the molecular basis of sex differences in this causal relationship, we examined shared loci that associated with testosterone and high blood pressure in each sex. In females, six loci ($P < 5 \times 10^{-8}$) were shared between testosterone and high blood pressure, and 3 (*CYP11B1*, *SLC16A1*, *SLC22A7*) of them were in DMET gene regions (Fig. 3e). In males, only 4 loci were shared between testosterone and high blood pressure, none of these loci were located in the DMET gene regions (Supplementary Fig. 12). Variant rs7003319 in *CYP11B1* 3′UTR is associated with both testosterone level and high blood pressure in females, but not males. Colocalization analysis of these two traits revealed a shared genetic basis only in females (PPH4$_{females}$ = 0.73, PPH4$_{males}$ = 0.00034; Fig. 3f). *CYP11B1*-encoded enzymes catalyze glucocorticoid production and lead to the production of 11-ketotestosterone (11-KT)[38]. A previous study reported that 11β-hydroxylase deficiency (11βOHD), a rare autosomal recessive disorder caused by mutations in the *CYP11B1* gene, is associated with hypertension[39]. Further investigation of the interplay between *CYP11B1*, testosterone, and high blood pressure in the context of biological sex is warranted.

## Sex differences in DMET gene expression are linked to sex differences in drug response

Elimination of drugs and exogenous toxins is the major function of DMET gene products. Such processes mainly take place in the liver. Differences in the abundance and activity of DMET have been shown to play a major role in drug responses and side effects[10]. Although sex differences in the gene expression of DMET genes have been studied in different contexts[40], the impact of these differentially expressed genes on responses to relevant medications is sparsely reported. Here, we hypothesized that sex differences in DMET gene expression would

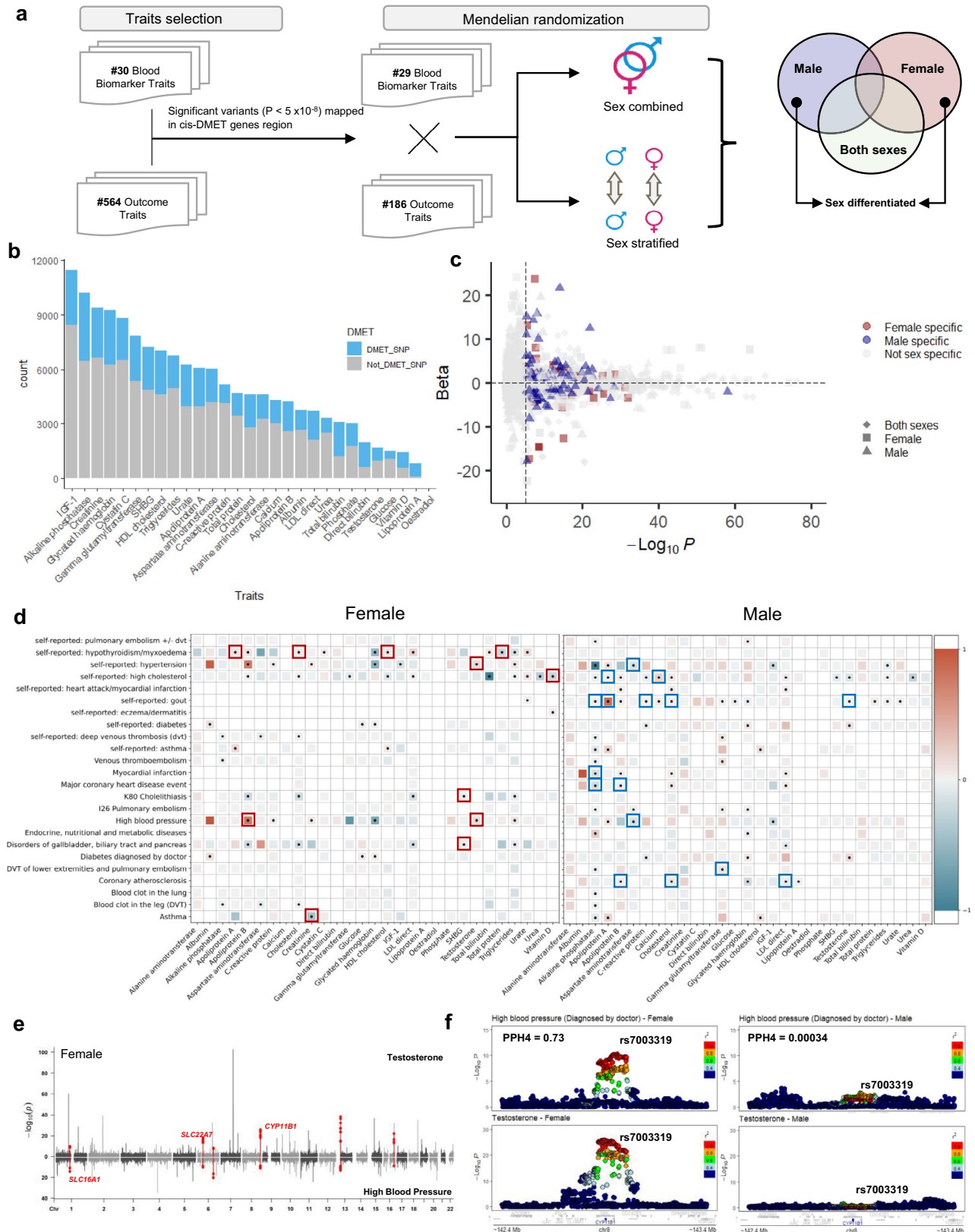

provide insights into observed sex differences in drug responses and side effects.

Using RNA-seq data from GTEx liver tissue, we identified 20 DMET genes with significant sex differences in expression; 10 expressed higher in males and 10 others expressed higher in females (FDR < 0.05, Fig. 4a, Supplementary Data 15). Among them, several well-

characterized pharmacogenes, such as *CYP1A2*, *CYP3A4* and *CYP2C19*, were found to exhibit sex-differential expression. When assessing the reproducibility of our discovery in an independent dataset, we recapitulated the differential expression for 14 of 19 genes (expression of *CYP1A2* was not quantified in the validation dataset) (Supplementary Fig. 13)[41]. Two additional smaller datasets[42, 43] were also evaluated and

**Fig. 3 | Sex-specific relationships between serum biomarkers and human health outcomes. a** Analytical pipeline for the Mendelian Randomization (MR) analysis to establish the causal relationship between serum biomarkers and human health outcomes. We evaluated SNP-trait associations for 30 serum biomarkers and 564 human complex traits from UKBB separately, focusing on those SNPs in DMET gene regions. MR tests were conducted using both sex-combined and sex-stratified GWAS summary statistics, limiting to those traits with significant DMET region genetic associations ($P < 5 \times 10^{-8}$), which include 29 exposures (serum biomarkers) and 186 outcomes (traits). **b** SNPs in DMET gene regions are important in the genetic regulation of serum biomarkers from UKBB. The barplot shows the total count of significant variants in the sex-combined serum biomarker analysis. The proportion of variants in DMET gene regions is shown in blue. **c** Volcano plot of sex-combined and sex-stratified MR results. The *x*-axis represents the $-\log_{10}$ (*P* value) from the MR-Egger test. The *y*-axis represents the effect size of MR-Egger tests. Each point represents a MR test from sex-combined (diamond) or sex-stratified (square if females; triangle if males) GWAS summary statistics. The color of points indicates sex-specificity of the MR relationship; blue indicates males (blue) or females (red) or not sex specific (gray). The plot does not display those cases with extremely large $-\log_{10}(p)$ values without sex-specific MR relationships. The vertical dashed line represents the significance threshold ($P < 0.05/(186*29)$). **d** Sex-stratified MR results between 24 diseases and serum biomarkers. Rows represent 24 diseases as outcome and columns represent 28 serum biomarker traits as exposure. The color represents the effect size. Significant causal effects are indicated with a solid black dot. Colored outlines highlight sex-specific MR relationships, with female-specific in red, and male-specific in blue. **e** Manhattan plot for GWAS of testosterone (top) and doctor-diagnosed high blood pressure (bottom) in females. SNPs associated with both traits ($P < 5 \times 10^{-8}$) are highlighted in red. SNPs in DMET gene regions and included in the MR test are labeled with DMET gene names. **f** Colocalization between testosterone level and doctor-diagnosed high blood pressure in the *CYP11B1* locus, shown separately for males and females. The posterior probability of shared causal effects (PPH4) is calculated separately for each sex. Linkage disequilibrium between variants is quantified by the squared Pearson coefficient of correlation ($r^2$).

our top differential expressed genes, such as *UGT2B17, UGT2A3, CYP3A4, SLC3A1, SLC16A14* are concordant with previous finding (Supplementary Data 16). Protein abundance of two pharmacogenes, CYP1A2 and CYP3A4, was quantified in pooled human liver microsomes (HLMs) that were collected separately from male and female donors. We confirmed higher protein abundance of CYP1A2 in the male-pooled HLMs and the opposite trend for CYP3A4 (Fig. 4b), which is correlated with a previous report[11].

To evaluate the potential clinical consequences of these sex differentially expressed DMET genes with respect to medication use, we annotated these genes with FDA-approved drugs. Specifically, using data from PharmGKB[44] and DrugBanks[45], both of which have summarized gene and drug relationships from published literature, we found 1,166 drugs that have been linked to at least one of the sex differentially expressed genes identified in this study (Supplementary Fig. 14, Supplementary Data 17). Not surprisingly, *CYP3A4*, a cytochrome P450 family member which is responsible for metabolizing approximately 50% of drugs on the market, has been linked with the greatest number of drugs.

To further investigate whether these annotated drugs have been reported to have sex differences in response, we performed web scraping to comprehensively search literature in PubMed for evidence supporting sex differences in response to these annotated drugs. Using "sex difference" and drug names as our key search terms, we identified 2,340 journal articles that contained information for 306 drugs. Given the high intensity of inspecting all articles, we focused on those drugs that were annotated with *CYP1A2*, as a proof-of-concept. This resulted in 519 journal articles which were then manually inspected to determine whether sex differences in drug responses were reported. Among them, we identified 75 articles that reported sex differences in either efficacy, pharmacokinetics, or toxicity of 49 drugs that are metabolized by CYP1A2 (Fig. 4c, Supplementary Data 18). Fluoxetine, for example, was reported to have greater weight reduction in females than in males at the same dosage[46]. Furthermore, antipsychotic medications, such as clozapine and olanzapine, have been reported to have a slower elimination in females than males[47], which is in agreement with our findings that males have a higher expression of *CYP1A2* (Fig. 4d).

Our analysis identified a number of sex-differential drug responses that were supported by existing literature. For example, flunarizine, another CYP1A2 substrate, which is used in treating epilepsy[48], was reported to not affect catalepsy in male mice, but attenuated catalepsy in females at the same doses[49]. Another study found that male rats formed two oxidative metabolites of flunarizine at higher rate than female rats[50]. These are in agreement with our findings that higher CYP1A2 expression in male can lead to faster metabolism/breakdown of this drug and therefore less response. Similarly, female mice have been reported to be less susceptible to acetaminophen overdose induced hepatotoxicity than male mice[51]. Acetaminophen is also metabolized by CYP1A2 to N-acetyl-p-benzoquinone imine (NAPQI), known to induced hepatotoxicity. In this case, the lower expression of CYP1A2 in female liver would lead to lower production of the toxic metabolite and therefore partially explain the lower toxic response in females.

Given the wide usage of clozapine and the higher rate of clozapine-induced toxicity observed in males[52], we hypothesized that the higher abundance of CYP1A2 contributes to higher formation of an active metabolite, and subsequently leads to higher rate of clozapine-induced side effects in males (Fig. 4e). We quantified the sex differences in clozapine metabolism using separate pools of HLMs from men and women. Because CYP3A4 also contributes to the metabolism of clozapine, in our HLM experiments we employed ketoconazole, a CYP3A4 inhibitor, to specifically evaluate CYP1A2-mediated metabolite formation. In the presence of ketoconazole, the formation of N-demesthyl-clozapine, the active metabolite, was significantly higher in the male HLMs (Fig. 4f, $P = 0.0309$). We observed no sex differences in the formation of an inactive metabolite, clozapine N-oxide (Fig. 4f) and parent drug clozapine (Supplementary Fig. 15). Higher plasma concentration of N-desmethylclozapine has been associated with clozapine side effects, such as agranulocytosis[53]. Our observation supports the notion that the observed higher frequency of clozapine-induced toxicity in males can be due to elevated formation of N-desmethylclozapine resulting from higher CYP1A2 levels in males (Fig. 4e).

## Discussion

In this study, we comprehensively examined the genetic basis of DMET genes to over 500 human complex traits in a sex-aware fashion. Unlike the majority of previous genomic studies, sex-differentiated genetic effects and regulatory mechanisms were identified by sex-stratified analytical model. We performed various analyses including heritability estimation, colocalization, genetic correlation, MR to survey global and regional genetic basis with the goal of not only establishing correlation but also inferring potential causality. When a sex-differentiated SNP-trait association was observed, a number of biologically plausible hypotheses were tested for causality, which include the identification of sex-differentiated genetic regulation of gene expression (eQTL) and serum biomarker level.

We observed significant sex differences in genome-wide heritability estimated for 83 different human complex traits. When focusing on the DMET gene regions, we identified a number of SDEs that are associated with human complex traits. Using colocalization we identified 35 sex-specific causal loci corresponding to eight traits. The majority of these traits have reported sex differences in disease prevalence, such as gout, hypothyroidism, and major coronary heart disease[1]. For example, we discovered that rs2360872 in the upstream

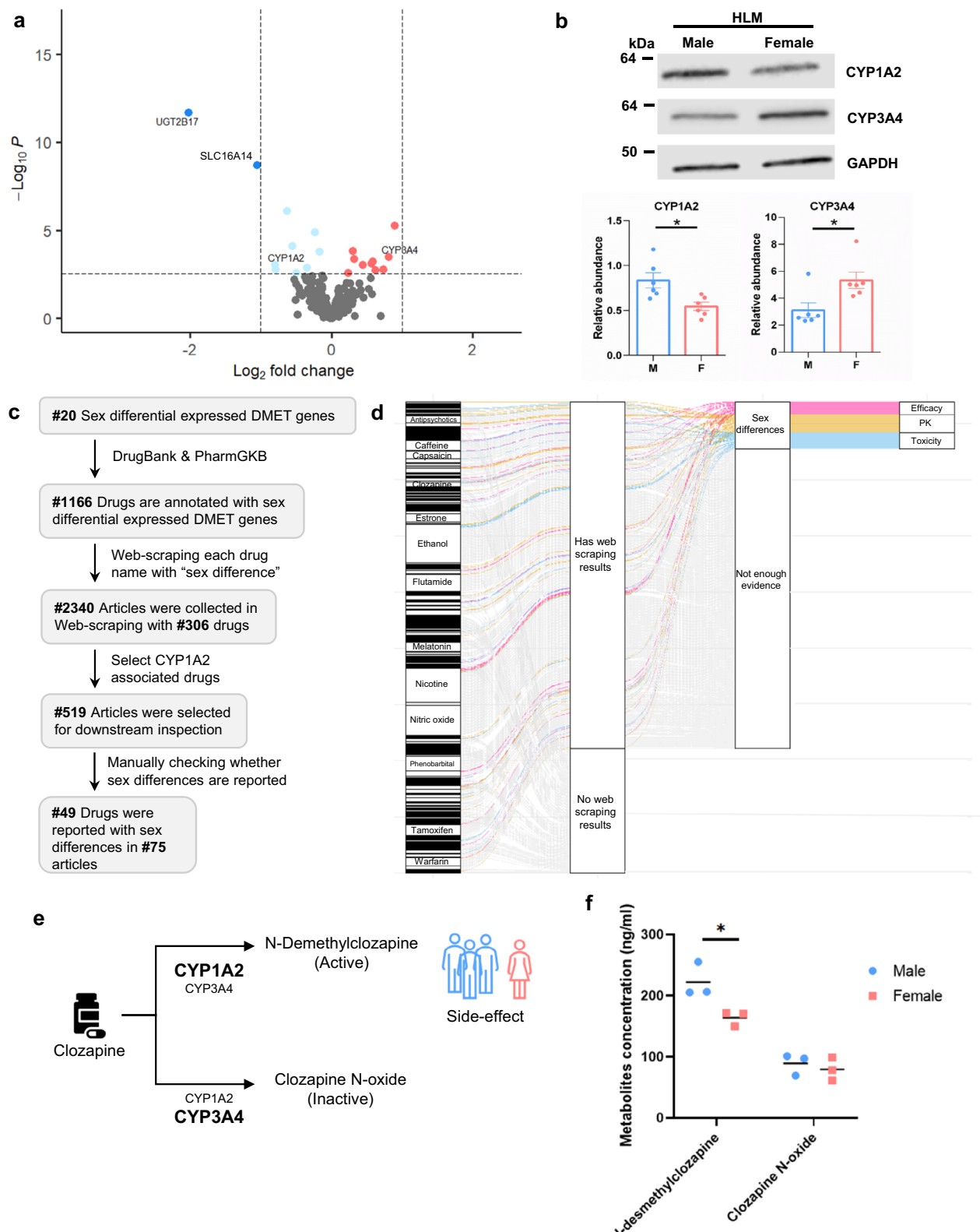

of urate transporter 1 is significantly associated with gout exclusively in males. Our findings highlight the need of incorporating sex as a biological variable in analysis approaches, especially for traits with known sex differences.

We characterized sex differences in genetic regulation of gene expression of DMET genes. Previous GTEx pan-tissue study identified few sex-biased expression quantitative trait loci (sb-eQTLs)[8]. Only 3 sb-eQTLs ($q < 0.25$) were reported in human liver, where none of them are

mapped into DMET gene regions. The design of the prior GTEx study was to provide a genome-wide characterization of sex-biased genetic regulation across tissues. As such, due to the high multiple testing burden, the study was powered only to detect large effects. In the present study by restricting the genes of interests to DMET genes, we alleviate the multiple testing correction penalty. Specifically, we detected sex-differentiated/specific *cis*-regulation of expression of six DMET genes in human liver. Male-specific regulations were detected

**Fig. 4 | Sex differential DMET gene expression is linked to sex difference of clinical drug response. a** Volcano plot of differential gene expression of DMET genes in human liver. Each point represents a DMET gene. There are 10 DMET genes more highly expressed in males (blue), and 10 more highly expressed in females (red). Wald test was performed using DEseq2. The horizontal dashed line represents FDR < 0.05. **b** Western blot analysis of sex-stratified human liver microsomes (HLMs; female HLMs were pooled from 21 donors, male HLMs were pooled from 12 donors) showing expression of CYP1A2 and CYP3A4. Top: Representative Western blot. Bottom: Quantification of CYP1A2 and CYP3A4 protein normalized over GAPDH from six replicates. Data are presented as means ± SEM. $P_{CYP1A2}$ = 0.0126, $P_{CYP3A4}$ = 0.0209. **c** Workflow for web-scraping analysis. **d** Web-scraping identified 49 CYP1A2-associated drugs with reported sex differences in clinical drug response. Sanky plot shows the number of articles reporting sex differences in drug response for those drugs annotated with CYP1A2 in DrugBank and/or PharmGKB.

**e** Schematic representation of sex difference in clozapine metabolism and toxicity. The active metabolites (N-desmethylclozapine) were primarily produced by CYP1A2, which has a higher abundance in the male liver. More N-desmethylclozapine could contribute to the reported higher frequency of side effects in males receiving clozapine. Font size of the gene name represents the relative contribution of each gene in generating the corresponding metabolites. **f** Impact of sex on clozapine metabolites generated in male and female pools of HLMs. N-desmethylclozapine, an active metabolite of clozapine and clozapine N-oxide, an inactive metabolite of clozapine, were shown in (**f**). Metabolite concentrations were measured by liquid chromatography-mass spectrometry (LC-MS). Each point represents an independent microsomal incubation experiment. The horizontal bar indicates the mean concentration. The color of points indicates sex (Blue: male; Red: female). *P* values (*P < 0.05) were determined by two-tailed *t* test.

for genes *PON1* and *UGT2B17*, which provide putative explanations for reported sex differences in the abundance of protein[28]. Our work is complementary to the previous GTEx global analysis. The present study highlights the need and potential to reveal sex differences in the genetic regulation of gene expression by focusing on a collection of important functional genes. The results from our work provide plausible mechanistic explanations for the observed sex differences in several human health traits.

The main function of DMET genes is to transform endogenous and exogenous molecules[10,15]. Indeed, we observed a large proportion of serum biomarker-associated SNPs were mapped into DMET gene regions in comparison to randomly selected SNPs, which further confirmed the critical role of DMET genes in the genetic basis of serum biomarkers. Further, sex-differentiated genetic effects on the level of serum biomarkers may also provide explanations for observed sex differences in higher-order complex traits. Consistent with previous reports[34,54], we found that hormone-related traits harbor the most sex differences in genetic architecture. By conducting sex-stratified MR, we revealed hundreds of sex-specific serum biomarker and traits associations, which were not identified in sex-combined models. Interestingly, on average, 40% of these causal relationships remain significant when only including the DMET SNPs as instrumental variables. Of note, these sex-differentiated causal inferences can represent different magnitudes of causal effect, different causal SNPs, or even differences in pleiotropic effect between males and females. Overall, we demonstrated sex differences in the genetic basis of serum biomarkers and a role for sex-differentiated effect of DMET gene in regulating those endogenous compounds.

Lastly, we evaluated sex differences in the expression of DMET genes. We identified 20 DMET genes with significant sex-differentiated expression. We further confirmed the sex differences in the protein abundance of CYP1A2 and CYP3A4 in HLMs. Because *CYP1A2* expression is highly inducible by smoking[55], we carefully designed our protein quantification experiments to include the same percentage of smokers in each separate male and female donor pool. To further explore the clinical impact of our findings, we extensively examined published literature of sex differences in CYP1A2 metabolized drugs. Of 519 articles, we found 49 drugs with previously reported sex differences in the PK profile, efficacy or toxicity. Among them, clozapine and olanzapine showed a consistent higher rate of adverse events in males[52]. We investigated whether this observation could be due to sex differences in metabolizing clozapine and found that a higher level of the active metabolite, N-desmethylclozapine, was detected in male-pooled HLMs, suggesting that male HLMs have higher CYP1A2 abundance. Both clozapine and N-desmethylclozapine are considered efficacious in treating schizophrenia and inducing side effects[56]. However, N-desmethylclozapine has a longer half-life than clozapine[57]. We proposed that a higher abundance of CYP1A2 in males leads to the accumulation of N-desmethylclozapine, which consequently increases the likelihood of

drug side effects. These results suggest that sex differences in DMET gene expression could affect the drug response.

Our study has limitations. First, our analyses were performed in the DMET gene regions (with 1 Mb flanking) rather than only the coding region of DMET genes. Our rationale is that the proximal gene region contains regulatory elements, such as promoters and enhancers, and genetic variants in these regions can have large impacts on gene expression[25]. By this definition, situations can rise where sex-differential genetic effects are actually coming from variants mapped onto nearby non-DMET genes. Therefore, one needs to be careful when interpreting the results from our work and avoiding simply calling DMET gene regions as DMET genes. Second, sex differences in genetic effects could be confounded by both sociological and behavioral differences between males and females. Such differences complicate the detection of true molecular mechanisms of disease. We noted that the potential sex differences highlighted in our results need to be distinguished with many features of behavior and external environments in future investigations. Third, for traits that exhibit sex differences in incidences (e.g., hypothyroidism), there is often imbalance in the sample size between the two sexes, which could affect the power of sex-stratified GWAS discovery. This is partly why we only selected traits that have cases number > 300 in both sexes for our analysis to avoid simply missing findings due to small sample size in one sex. We acknowledge that this might not be enough. In an ideal scenario, GWAS could be performed after matching the sample size between males and females. However, having access to only the summary statistics from the UKBB prevented us from taking this approach. Last, our analyses were built upon publicly available GWAS summary statistics which were generated using a simple linear regression model. However, this approach has the potential to produce spurious results when dealing with case/control traits where the ratio of cases number to control number is high[58]. This impact could be severe in sex-stratified GWAS analysis. Methods such as the Logistic Mix Model (LogMM) can account for case/control imbalance[59]. We therefore re-ran our analysis using GWAS summary statistics from LogMM for 11 traits and compared it with our original discovery. We found that results were highly consistent with our original analyses. For sex-stratified heritability estimation, we observed remarkable concordance between two methods ($Cor_{female}$ = 0.98, $P_{female}$ = 4.16 ×$10^{-8}$; $Cor_{male}$ = 0.63, $P_{male}$ = 0.039, Supplementary Fig. 16, Supplementary Data 19). We observed similar results for estimates of male-female genetic correlation (Supplementary Fig. 17, Supplementary Data 19) and the number of SDEs (Supplementary Fig. 18 & Supplementary Fig. 19, Supplementary Data 20). There is only one exception: the self-reported heart/cardiac problem trait. For this particular trait, the estimated heritability and the number of SDEs changed between the two methods in males. Manhattan plots showed that more trait-associated SNPs ($P < 5 \times 10^{-8}$) have been identified using LogMM compared to regular linear regression (Supplementary Fig. 20). Of note, this difference might not only be resulted from the analytical model, but could also come from the inclusion of participants, the selection of SNPs (e.g., MAF cutoff), and covariates in the designed

equation. In conclusion, we believe that our original discovery is similar to the results using LogMM method, and thus our results accurately reflect the sex differences of genetic effects in the DMET genes region.

In summary, we identified a number of human complex traits that were under different genetic regulation between the two sexes at both global and regional scale (focusing on DMET gene regions). Our results strongly support that sex-stratified analysis is critical for studying the underlying mechanisms of sex differences in human health. Using DMET genes as a gateway, we highlighted the translational impact of understanding sex differences in the genetic contribution to gene expression, endogenous and exogenous substrate formation. Critically, follow up studies based off sex-specific discoveries made from our work can and will improve our understanding of disease etiology and development and facilitate disease prevention and treatment.

## Methods

### Data collection

We obtained GWAS sex-combined and sex-stratified GWAS summary statistics from the UK biobank (UKBB) (release 2; http://www.nealelab.is/uk-biobank). Traits were selected based on 3 criteria: (1) We defined the genomic region for each DMET gene as the region 1 Mb up/downstream of the transcription start site (TSS) of each DMET gene of interest. Using the European Bioinformatic Institute (EBI) GWAS catalog (https://www.ebi.ac.uk/gwas/), we first selected those traits with trait-associated SNPs mapping to DMET gene regions, which resulted in 3100 unique studies. By manually extracting the keywords of each trait from the GWAS catalog and matching them to the phenotype description of UKBB GWAS summary statistics, we had 1105 traits (both binary/categorical and continuous) for subsequent analysis. (2) We removed categorical/binary traits with a) fewer than 300 cases in either sex, and b) no sex difference in prevalence (ratio of prevalence > 0.02)[60]. This resulted in 421 categorical/binary traits. (3) 30 serum biomarker traits were removed from initial analysis to be used for subsequent causality inference. In total we used GWAS data for 564 traits (421 categorical/binary traits and 143 quantitative traits) in the present study (Fig. 1a). For these traits, we obtained LDSC-estimated heritability (https://nealelab.github.io/UKBB_ldsc/index.html). For continuous traits, we selected the GWAS summary statistics where the phenotype was inverse rank normalized (IRNT). The variant effect and Combined Annotation Dependent Depletion (CADD) score were obtained from https://gnomad.broadinstitute.org/. Data analysis was conducted in RStudio 3.6.3.

Transcriptome data from human liver tissue were obtained from the NIH Genotype-Tissue Expression (GTEx) Portal (https://gtexportal.org/home/datasets) and Gene Expression Omnibus (GEO) repositories (GEO: GSE24293). All GTEx genotype data were obtained from dbGap (phs000424.v8).

Drug substrate information and clinical annotation for DMET genes were obtained from DrugBank (https://go.drugbank.com/releases/latest, release on 2021-01-03) and PharmGKB (https://www.pharmgkb.org/downloads, release on 2021-05-05). The PharmGKB clinical annotation consists of evidence from the literature of association between specific genetic variants and drugs. The DrugBank dataset recorded literature reported enzymes/genes that involve the metabolic process of the corresponding drug.

### Estimation of heritability and genetic correlation

Heritability and genetic correlation were calculated for each trait using the "HDL" R package, which utilized GWAS summary statistics and has been shown to improve precision in estimating genetic correlation than linkage disequilibrium score regression (LDSC)[61]. Briefly, because our data were from UKBB, we used pre-computed SNP panels available in the HDL package for the European-ancestry population, which were imputed to Haplotype Reference Consortium (HRC) and

UK10K + 1000 Genomes. We excluded SNPs mapping to the major histocompatibility complex (MHC) region or minor allele frequency (MAF) < 5%, resulting in a total of 1,029,876 SNPs passing QC. Detailed information about the reference panels is described in https://github.com/zhenin/HDL.

To quantify sex differences in estimated heritability, the z-score test to each of the 564 traits:

$$z - \text{score} = \frac{\text{STAT}_{\text{female}} - \text{STAT}_{\text{male}}}{\sqrt{\text{SE}_{\text{female}}^2 + \text{SE}_{\text{male}}^2}} \tag{1}$$

where STAT represents heritability and SE represents the standard error of the estimated heritability. Two-tailed $P$ values were calculated, and we controlled for multiple testing using the Benjamini-Hochberg FDR approach. We defined statistical significance as FDR < 0.05.

To test whether genetic correlation between males and females differs from 1, we used the t-score test:

$$t - \text{score} = \frac{r_g - 1}{\text{SE}_{rg}} \tag{2}$$

where $r_g$ represents the genetic correlation coefficient and the $\text{SE}_{rg}$ represents the standard error. Two-tailed $P$ values were calculated, significance determined by FDR < 0.05.

### Sex-differentiated genetic effects

To quantify sex differences in gene-trait associations for those SNPs in the DMET genes regions, we used Eq.(1) to test for sex differences in male and female genetic effects for all SNP-trait pairs. In this case, the STAT is the GWAS effect size (β) from sex-stratified GWAS, and SE is the standard error of the effect size. Two-tailed $P$ values were calculated. We defined SNPs with sex-Differentiated Effects (SDEs) as those with significant sex differences in genetic effects (FDR < 0.05) and significantly associated ($P < 5 \times 10^{-8}$) with the trait in at least one sex.

### Two-sample Mendelian Randomization analyses

To test whether sex-differentiated genetic basis on serum biomarker levels might mediate sex-differentiated casual relationships, we applied mendelian randomization (MR) to a total of 29 serum biomarker traits as exposures and 186 human complex traits as outcomes that were selected from the aforementioned 564 human complex traits. In brief, the traits were selected based on whether there are shared significant variants ($P < 5 \times 10^{-8}$) between exposure and outcome that are mapped into the DMET genes region. We applied MR-Egger regression to estimate the causal effect of genetically regulated serum biomarker levels on outcomes. The MR tests were performed using sex-combined and sex-stratified analyses. We applied a stringent statistical threshold (0.05/(186*29); 186 outcomes and 29 exposures) to define statistically significant MR relationships. A sex-specific MR relationship was defined as MR significant in a single sex, but not in the other sex and sex-combined model MR test. All MR analyses were performed using the "TwoSampleMR" R package[62].

### Sex-stratified cis-eQTL analysis in human liver

To quantify the association between genotype and gene expression, we obtained summary statistics of sex-stratified cis-eQTL in human liver from the GTEx project. In brief, a linear regression model was applied using FastQTL while adjusting for additional and unknown factors as previously described in[8].

$$Y \sim \beta_0 + \beta_G \text{Genotype} + \beta_{(1\ldots m)}C + \varepsilon \tag{3}$$

Where $Y$ is the gene expression, $\beta_O$ represents the intercept, $C$ represents covariates that were used in cis-eQTL mapping. The total number of male liver samples was down-sampled to match the female sample

size for maintaining the same discovery power. We applied a 2-step multiple testing correction: (1) using Bonferroni correction to account for the number of independent SNPs tested within each DMET gene region; (2) using FDR to account for the number of tested genes. We defined independent SNPs in each region as those with $r^2 < 0.8$ with the SNP in the same region identified using the *"LDlinkR"* package[63]. Linkage disequilibrium information was calculated using the "CEU" (Utah Residents (CEPH) with Northern and Western European ancestry) population as a reference. FDR < 0.1 was defined as a statistically significant threshold for sex-stratified *cis*-eQTL. To identify sex-differentiated eQTL, we once again applied the z-score test from Eq. (1). FDR < 0.1 is used to define sex-differentiated eQTL. In addition, eQTL analysis was conducted in males and females separately. We identified sex-specific eQTLs that is statistically significant in only one sex (FDR < 0.1) but not statistically significant in the other sex (FDR > 0.1).

### Sex-stratified colocalization analysis

For SDEs and those eQTLs with evidence of sex-differentiated/specific effects, we performed colocalization analysis using the *"coloc"* R package[21]. For SDEs, we performed colocalization analyses of 12 traits using GWAS summary statistics between sexes. The genomic regions for colocalization are defined as ±200k base pairs from the tagging SDEs ($r^2 < 0.2$). For case-control traits, the effect size and their variances were used. For quantitative traits, the standard deviation of the measure outcome is not available. We input the MAF and sample size that are recorded in each GWAS summary statistics to run the coloc.abf() function. The PPH1 indicates the posterior probability that only males have a genetic association in the testing region. The PPH2 indicates the posterior probability that only females have a genetic association in the testing region. The PPH3 indicates the posterior probability that both sexes have a genetic association in the testing region; however, the causal variants are different. The PPH4 indicates the posterior probability that male and female share the same causal variant.

For colocalization between sex-combined GWAS summary statistics and sex-stratified eQTL, we applied coloc.abf() to all variants in the gene region of each sex-differentiated/specific *cis*-DMET eQTLs (±1 Mb) that were available for both *cis*-eQTLs and GWAS. The input for GWAS summary statistics and eQTLs were the same as previous descriptions. The PPH4 indicates the posterior probability that the traits share the same causal variant with either female or male-specific eQTLs. We defined sex-specific colocalization as those with PPH4 larger than 0.5 in one sex but not the other.

### Identification of sex-differentiated DMET gene expression in human liver

We quantified sex differences in DMET gene expression in GTEx liver tissue using the *DEseq2*[64]. The list of DMET genes was obtained from[10], and filtered to remove low-expressed genes and genes that are on Y chromosomes and mitochondria. Retained genes are those where at least 20% of samples have gene count larger than six and TPM larger than 0.1. We then fit a generalized linear model to each gene to test for sex-differentiated expression using the *'DEseq2'* package, while controlling for known sample characteristics such as ischemic time and RNA integrity (RIN), and 13 surrogate variables (SVs), which were defined by the *"smartSVA"* package[65]. The SVs were estimated from the whole transcriptome after removing low expressed genes that capture confounding effects from technical and biological factors. Differentially expressed genes were defined as those with Benjamini-Hochberg FDR < 0.05.

To validate the differentially expressed DMET genes, we characterized sex differences in gene expression in an independent dataset comprising microarray data from healthy liver tissue (GSE24293)[41]. For those DMET genes differentially expressed in GTEx liver, we assessed differential expression in the replication dataset using Welch's t-test,

defining significant replication as those DMET genes with *P* value <0.05 in the second dataset.

### Web scraping for literature evidence of sex differences

To establish links between the sex differentially expressed DMET genes and clinical treatment outcomes, we applied web scraping using *"easyPubMed"* (https://cran.r-project.org/web/packages/easyPubMed/index.html). Drugs that that are metabolized by the sex-differentially expressed DMET genes were obtained from PharmGKB and DrugBank. We searched PubMed.gov for articles with the drug names and "sex difference" in the title and abstract. This resulted in a total of 2340 unique publications. After excluding articles primary focused on sex differences in endogenous sex hormones, such as testosterone and estradiol, 519 articles remained. We manually inspected each article to determine if the article reported sex differences in drug efficacy, toxicity, or pharmacokinetics (PK) based on the following criteria: (1) the drug demonstrated sex differences in pharmacokinetics; (2) for Food and Drug Administration (FDA) approved indications, the drug displays sex differences in efficacy; (3) the drug displays sex differences in frequency of drug-related toxicity/adverse effects; (4) the drug did not serve as an inducer/inhibitor of other drugs in the study.

### Protein quantification in human liver microsome (HLM)

Male-pooled HLM from 12 male donors (Corning, Catalog: 452172, Lot: 1077002) and female-pooled HLM from 21 donors (Corning, Catalog: 452183, Lot: 5061002) were lysed in the RIPA buffer. Protein concentrations were then determined using a Bicinchoninic acid (BCA) assay. Samples were then denatured, resolved in Tris-glycine gel, and blotted with the primary (ABclonal, CYP1A2: A0062, CYP3A4: A2544, GAPDH: A19056, 1:1000) and secondary (ABclonal AS014, 1:20000) antibodies.

### Clozapine metabolism in single-sex HLM pools

The experiment was performed as described[66]. Briefly, 0.4 mg male or female pooled HLM was incubated with 100 μM clozapine with the presence or absence of 2 μM ketoconazole (CYP3A4 inhibitor). After 30 min of incubation, the reaction was stopped by adding ice cold acetonitrile. Protein was precipitated twice, and the collected supernatant was stored at −20 °C. Lastly, a 10 μl portion of the supernatant was injected into the LC-MS system to determine the concentration of parent compound and metabolites[67]. The detection and quantification of clozapine and metabolites was performed using high-performance liquid chromatography (Agilent 1200 Series, Santa Clara CA) coupled with mass spectrometry (TSQ Quantum triple stage quadrupole mass spectrometer; Thermo-Electron, San Jose, CA). Chromatographic separation was performed with a Phenomenex Polar RP column, 75 × 2.0 mm, 4.0 micron, (Torrance, CA) with a mobile phase containing (60:40) DI water with 0.1% formic acid: Acetonitrile with 0.1% formic acid, at a flow rate of 500 μL/min, with the column temperature set at 30 °C. The experiment was repeated independently 3 times. Data acquisition was performed with Xcalibur Version 2.07 (Thermo-Electron). Statistical analyses were performed using the student *t* test. *P* value <0.05 was defined as a statistically significant threshold.

### Reporting summary

Further information on research design is available in the Nature Portfolio Reporting Summary linked to this article.

## Data availability

The UKBB GWAS summary statistics by the Neale laboratory can be obtained from http://www.nealelab.is/uk-biobank/. The summary statistics of cis-eQTL is available at the GTEx (https://gtexportal.org/home/). All GTEx protected data are available via dbGaP

(phs000424.v8). Differential gene expression validation dataset is available at GSE24293. Drug substrate information and clinical annotation for DMET genes were obtained from DrugBank (https://go.drugbank.com/releases/latest, release on 2021-01-03) and PharmGKB (https://www.pharmgkb.org/downloads, release on 2021-05-05). The LDSC-estimated heritability (https://nealelab.github.io/UKBB_ldsc/index.html). The variant effect and Combined Annotation Dependent Depletion (CADD) score were obtained from https://gnomad.broadinstitute.org/. Source data are provided with this paper.

## Code availability

HDL software is available at https://github.com/zhenin/HDL/. The code used in this manuscript is available at Open Science Framework (OSF) and are stored in the "Decipher genetic underlying causes for sex differences in human health through the lens of drug metabolism and transporter genes" project, which can be accessed at https://osf.io/vfpjx/.

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

## Acknowledgements

This study is supported by NIH/NCI Grants R01CA229618 (to R.S.H. and B.E.S.) and NIH Grants R01HG011405 (to B.E.S.). We thank the donors and their families for their generous gifts of biospecimens to the GTEx research project. We thank Dr. James Fisher in Clinical Pharmacology Analytical Services (CPAS) for his help in developing the LC-MS assay. We thank individuals at the University of Minnesota for technical support in executing experiments. We thank the 3M Science and Technology Fellowship, Bighley Graduate Fellowship, A-PReP Scholarship (UMN-CTSI) for their support.

## Author contributions

Conceptualization: Y.H., R.S.H.; Methodology: Y.H., Y.S., W.Z., F.L., A.M.L., B.E.S., R.S.H.; Investigation: Y.H., Y.S., W.Z.; Visualization: Y.H., W.Z.; Funding acquisition: B.E.S., R.S.H.; Project administration: A.M.L., R.S.H.; Supervision: B.E.S., R.S.H.; Writing—original draft: Y.H., R.S.H.; Writing—review & editing: Y.H., Y.S., W.Z., F.L., A.M.L., B.E.S., R.S.H.

## Competing interests

The authors declare no competing interests.
