## [Peer Review File · Nature Communications]

Deciphering genetic causes for sex differences in human health through drug metabolism and transporter genesREVIEWER COMMENTS

Reviewer #1 (Remarks to the Author):

This study aims to identify sex-differentiated genetic factors that influence drug metabolism using genomic, transcriptomic and phenotypic data that have been deposited in databases such as the UK Biobank and GTEx. The study uses multiple state-of-the-art statistical analysis techniques to evaluate sex differences at multiple levels. The study addresses a significant and understudied area. The findings generally do not provide functional characterization of the gene variants identified, but nevertheless may provide a useful resource for other investigators.

—From the outset, the authors focus their analysis on genes encoding proteins involved in drug metabolism and transport (referred to as DMET genes), as these are candidates that are known to influence metabolism of xenobiotic compounds. Apologies if I simply missed it, but searching through the copious supplementary data (17 tables and 18 figures), I could not find a list of these genes, nor an explanation of how the authors defined them, which processes they are involved in, etc. This needs to be included near the beginning of the manuscript.

—Related to the focus on DMET genes—although it is reasonable to assess DMET genes as a proof-of-principle, this negates the unique value of genome-wide studies in that they are agnostic. The authors ultimately did include analysis of biomarker associated-SNPs across the genome, which showed that DMET loci accounted for only a small proportion of these (Fig. 3B). Apologies if I missed it, but I could not find data for the non-DMET loci. Are these provided?

—The data shown in Fig. 4F represent the only experimental analysis to test a potential role of sex differences in drug metabolism gene expression and drug metabolites in the circulation of men and women. Obviously, these are difficult studies to perform, but given the large number of potential relationships uncovered here, additional functional tests would dramatically strengthen the impact of the analyses that were performed.

Comments on presentation of the data:

The text in many of the figure panels is extremely small and not accessible to the reader. Particularly frustrating is the illegibility of labels for the groups along both the x and y axes of key data summary figures such as Fig. 1E, 2C, and 3D. These are nicely conceived schemes to represent complex data, but are illegible without magnifying to 200%.

The text could benefit from editing to ensure that meaning is conveyed as clearly as possible. For example, the sentences at lines 66–70 are not stated well (do the authors mean "...DMET genes are not limited to mediating the blood concentration of xenobiotics, but also determine the amount of....a comprehensive study of sex differences in DMET gene activity and their health impact..."?). Other instances occur throughout the text.

Reviewer #2 (Remarks to the Author):

Really important lens of looking at sex differences through DMET genes -- and I think this is an excellent way to approach examining sex-specific genetic effects and is an exciting lens. I appreciated the thoroughness of genetic analysis methods used (heritability, GWAS, eQTL, expression, MR) and the follow up in liver microsomes. However, I think the lens could benefit from additional justification (by comparison with non-DMET regions), and the sex differences results require further comparison to

the literature and acknowledgement of potential contributing covariates.

Justification of the DMET lens and comparison to overall results

Particularly, in results section 1 - you examine heritability of traits with at least one significant DMET region SNP.

- (1) How many traits did you start with?
- (2) How was the set of DMET genes selected? What is defined as a DMET region?
- (3) What is the reasoning behind filtering traits in this way? Would it not make more sense to estimate sex-specific DMET region heritability vs overall heritability of traits that have a DMET SNP?
- (4) How does the metric for sex differences in heritability among these traits (14.7%) compare to that for 564 randomly selected traits (that don't have a DMET region SNP)? This comparison with background would justify the lens you're using.
- (5) Could you partition the heritability of traits into DMET region vs non-DMET region? I think this could yield additional insights

The same is true for the DMET lens in general -- e.g. line 169 you mention you characterized sex differences at a genome-wide and in DMET regions, but I see no mention of the genome-wide examination or comparison with the DMET region results. I also want to see this justification for the MR analysis.

eQTL analysis -- how many eQTLs did you examine for sex-diff and sex-spec effects

More acknowledgement of results to date in liver GEX studies

The literature search does not add much to the paper, instead a comparison to previously reported sex differences in liver expression would improve the results.

- multiple papers on sex-diff DMET genes (e.g. Yang L, Li Y, Hong H, Chang CW, Guo LW, Lyn-Cook B, Shi L, Ning B. Sex Differences in the Expression of Drug-Metabolizing and Transporter Genes in Human Liver. *J Drug Metab Toxicol.* 2012; Zhang Y, Klein K, Sugathan A, Nassery N, Dombkowski A, Zanger UM, Waxman DJ. Transcriptional profiling of human liver identifies sex-biased genes associated with polygenic dyslipidemia and coronary artery disease. *PLoS One.* 2011) -- how do your results compare?

- the CYPs examined (1A2 and 3A4) in HLMs have known sex differences in expression in liver (you cite Waxman and Holloway earlier, but this should be acknowledged when you do the examination

Acknowledgement of non-genetic sex-gender differences that may contribute to results

- Many of the SDE traits that come up (hypothyroidism, gout) in sex differential heritability are (1) self-reported (there are known gender differences in reporting behaviors) and (2) have known sex differences in incidence (hypothyroidism is more common in women, gout in men). How do you account for this?

- in discussion of alcohol intake locus (1221-234) -- there are known sex-gender differences in alcohol consumption that are not necessarily genetic, and relate to body size and sociocultural patterns, make sure to acknowledge this

- Same is true re incidence and behaviors for many of the traits mentioned in figure 2

MR analysis needs more examination

For the MR analysis, it is not clear which SNPs are used as instruments. It is important that the instruments are not selected from the same GWAS as the MR analysis is performed on -- and it is not clear if this is the case. Also there are a couple cases where the results could be affected by winner's curse: first, "traits were selected based on whether there are shared significant variants between exposure and outcome that are mapped into the DMET genes region", and second, a follow up MR analysis was performed only looking at "SNPs in DMET regions in traits with sex-specific causal effects". Follow up analysis in a validation cohort is required.

Minor notes:

The paper would benefit with clearer justification of bridge between DMET genes and other traits of interest (e.g. in the abstract -- mention high BP without making the jump to why we are looking at non-drug traits, remove and make this clearer in the abstract or intro)

Clearer distinction between sex-differential and sex-specific effects - clear the authors understand, but define this earlier (e.g. introduction). Also for places where you just mention the sex-specific effects (e.g. causal loci line 147), also test for differences.

- title: I would change "decipher" to "deciphering"
- abstract could be clearer, I found it hard to follow (e.g. remove "For example" line 20)
- introduction lines 63-66 -- should acknowledge that PGx studies often do not consider sex because of size/power limitations (and generally acknowledge this as a problem for sex-separated or sex-aware GWAS)
- line 83 -- multiple previous papers on this! cite them
- line 90 -- sex differences in genetic architecture *of DMET genes* <-- this is what you were looking at
- line 137 "However" sentence is not a complete sentence
- Figure 1 -- could not read the figure, resolution is too low
in part D: "traits show" not shows
- what are sex heterogeneity SNPs? define this
- line 223 "Therefore" - this does not directly follow and requires a citation
- Supplemental Table legends - please describe the columns in more detail in the "Meta" sheet or elsewhere.
- line 253 citation for sex differences in serum biomarkers is about testosterone specifically, include other citations that describe these differences
- line 372: PharmGKB and DrugBank require citations
- I think you can add more lead up to the microrosome analysis -- this is a strength of the paper but was hard to follow when mentioned on line 365-368. Mention that you did this as a follow up analysis. May want to move to another section
- line 383: "high intensity" is not the correct word
- line 480-481: I am not sure what you are referring to here?

Point-by-point Responses to the Reviewers' Comments

NCOMMS-22-24647

Original Title: Decipher genetic underlying causes for sex differences in human health through the lens of drug metabolism and transporter genes

We would like to thank the editor and the two reviewers for their critical review of our manuscript. It is nice and refreshing to receive comments from the reviewers who understand our work and put forward recommendations that are thoughtful and constructive. We are grateful for the recognition from both reviewers regarding the importance of our research topic and the significance of employing multiple statistical analysis techniques to examine the issue. As recognized by the reviewers, genetic underlying causes for sex differences in human health is an understudied area. To present answers to the fundamental question of whether and how genetic may contribute to sex differences in human health, we chose to focus on a collection of genes (encoding drug metabolism enzymes and transporters, DMET) and comprehensively evaluate their role in affecting sex differences in a wide range of human complex traits including drug response phenotypes. We systematically studied the genetic regulations of gene expression, serum biomarkers and drug responses in men and women separately and identified a number of sex-specific genetic regulations which could lead to the observed sex differences for a number of human health traits. The analytical pipeline employed in our study can be applicable to any additional genes/collection of genes of interest, beyond the DMET genes, or specific phenotypes of interest. Specifically in this revised manuscript, we have extended our analysis to compare results between DMET and non-DMET gene regions as suggested by reviewer 2. We have also provided additional evidence to support the functional validity of our pharmacogenomics discovery per reviewer 1's request. Please see our detailed responses to each reviewer's comments below.

Reviewer #1 (Remarks to the Author):

This study aims to identify sex-differentiated genetic factors that influence drug metabolism using genomic, transcriptomic and phenotypic data that have been deposited in databases such as the UK Biobank and GTEx. The study uses multiple state-of-the-art statistical analysis techniques to evaluate sex differences at multiple levels. The study addresses a significant and understudied area. The findings generally do not provide functional characterization of the gene variants identified, but nevertheless may provide a useful resource for other investigators.

Response: We thank the reviewer for the excellent summary of our work and appreciate the reviewer's recognition of our intention to provide a resource for other investigators through our comprehensive analysis of over 500 human complex traits. We agree with the reviewer that functional characterization of the gene variants identified is important. In our manuscript, we chose to functionally validate one of our discoveries between a key drug metabolism gene, *CYP1A2*, and clozapine metabolism separately in each sex. Given our group's expertise in pharmacogenomics, we are well prepared to conduct these experiments and have indeed validated our initial findings. However, given the broad range of health related phenotypes we examined and often the complex nature of disease etiology, we feel that it is more appropriate to

share our findings which will enable well designed and sophisticated experimental validation to be carried out by the broad research community. Nonetheless, we added additional literature evidence to sustain and support our discoveries throughout the revised manuscript.

—From the outset, the authors focus their analysis on genes encoding proteins involved in drug metabolism and transport (referred to as DMET genes), as these are candidates that are known to influence metabolism of xenobiotic compounds. Apologies if I simply missed it, but searching through the copious supplementary data (17 tables and 18 figures), I could not find a list of these genes, nor an explanation of how the authors defined them, which processes they are involved in, etc. This needs to be included near the beginning of the manuscript.

Response: The DMET gene list was retrieved from an existing publication ([10.1371/journal.pone.0060368](https://doi.org/10.1371/journal.pone.0060368)). We added a new Table S1 which provides details about these DMET genes. In addition, we also added content at the beginning of Results section as “DMET genes, which encode 222 metabolism enzymes and 150 transporters, were retrieved from a previous publication (Table S1).”

—Related to the focus on DMET genes—although it is reasonable to assess DMET genes as a proof-of-principle, this negates the unique value of genome-wide studies in that they are agnostic. The authors ultimately did include analysis of biomarker associated-SNPs across the genome, which showed that DMET loci accounted for only a small proportion of these (Fig. 3B). Apologies if I missed it, but I could not find data for the non-DMET loci. Are these provided?

Response: As stated above, when dealing with a wide-open topic with thousands of traits, tens of thousands of genes for potential evaluation, we chose to focus on a collection of DMET genes to comprehensively evaluate their sex-specific genetic regulation and health impact. The discovery pipeline employed in our study can serve as a roadmap to evaluate any other genes/pathways. Further, the smaller list of genes in our study will allow sufficient statistical power for our sex stratified analysis. Note, findings from our work should be interpreted in the context of chosen gene sets and phenotypes. For example, Fig 3B pointed out by the reviewer is focusing on 29 endogenous serum biomarkers and should be interpreted as the genetic variants within the DMET gene regions were associated with all 29 serum biomarker levels. Whether and how much these DMET region variants are important in other traits needs to be evaluated separately, as well as the relationship between these serum biomarker levels and any other genes of interest. We would argue that given only about 300ish DMET genes were evaluated among the possible tens of thousands of genes in the human genome, and significant DMET-SNP associations which account for between 5-95% of total significant associations with these 29 traits illustrated in Fig 3B, these presented solid evidence that it is important to study DMET gene variations for their potential regulation on serum biomarker level phenotypes.

To further justify the selection of DMET genes, we 1) estimated genome-wide h^2 in additional 1222 UKBB traits that are not included in our original manuscript; 2) estimated regional h^2 from DMET regions and non-DMET regions in a number of traits using the latest tool (Nature Genetic, [10.1038/s41588-021-00912-0](https://doi.org/10.1038/s41588-021-00912-0)). The rationale for the evaluation of these additional traits is that we only evaluated 564 traits that have been reported to relate to DMET region SNPs in our original manuscript. Now we expanded our analysis to traits that have not been reported to related DMET regions SNPs and quantitatively assessed to which degree sex differences is observed based on genome-wide heritability for these non-DMET related traits.

In the original manuscript, we reported 14.7% of traits (83/564) show sex differences in genome-wide h^2 . In the new analysis, we observed similar results that 13.7% of traits (167/1222) show sex differences in global h^2 (Figure below). These observations indicate 1) around 10%-15% of human complex traits in UKBB have sex differences in the fraction of the variance of a trait that is accounted for by genetic factors. 2) Our traits selection pipeline does not bias in selecting traits that might have higher or lower chances of sex differences.

We added these to the Results section as "When expanding the genome-wide heritability analysis to additional 1222 traits that have sufficient samples for both sexes in UKBB and are not known to be related to DMET genetic regions, we found 13.7% (167/1222) of them showing sex differences in global heritability (Fig. S2, Table S4). Interestingly, similar 13.40% (71/530) traits showing significant differences in their heritability between two sexes have been reported by an independent study."

Next, to gain a sense of the non-DMET genetic regulation for those traits that show sex differences in genetic regulation (filtered through genome-wide h^2 analysis), we estimated region specific h^2 using a latest tool (LAVA, Nature Genetic, 10.1038/s41588-021-00912-0) for both DMET and non-DMET regions for two traits: Gout and Hypothyroidism. We selected non-DMET regions that have relatively equal genomic length as the DMET regions when estimating their regional h^2 . As shown in the figures below, for Gout (top two figures), we observed DMET region h^2 (the blue dash line) fall within the distribution of a collection of randomly selected similar length non-DMET regions h^2 (represented by blue columns) in male, but a different pattern is observed in female with much lower DMET region h^2 (the red dash line) when compared to non-DMET regions h^2 . Unlike Gout, in Hypothyroidism (bottom two figures), we observed h^2 from DMET region is greater than non-DMET regions in female, but lower than non-DMET regions in male. Taken together, these observations support that for gout and hypothyroidism, global genetic regulation differs between the two sexes, and that the genetic effect of DMET regions impact each trait differently between the two sexes. These results also indicate that the sex differences in genetic effect is trait dependent. The genetic effects of DMET and non-DMET regions vary for different traits and should be interpreted only within the context

of that trait. We have carefully examined our manuscript to make sure that all interpretation of results were in the context of DMET genes and the qualitative and quantitative findings around this set of genes were not extended or generalized to other non-DMET genes.

—The data shown in Fig. 4F represent the only experimental analysis to test a potential role of sex differences in drug metabolism gene expression and drug metabolites in the circulation of men and women. Obviously, these are difficult studies to perform, but given the large number of potential relationships uncovered here, additional functional tests would dramatically strengthen the impact of the analyses that were performed.

Response: We thank the reviewer for recognizing the difficulty in validating genetic contribution to human complex traits. Genetic despite important, often only contribute to a portion of the cause of driving phenotypic variations. Among the hundreds of traits evaluated in our study, we viewed them generally as two types: disease/pathological related traits as well as drug response traits. Most of the human complex disease traits (aside from those Mendelian disorders), are known to be affected by large numbers of genetic variations in combination and are associated with genetic-environmental interaction. All of these pose questions in designing the right experiments in the right models for true validation. For these traits, our study will provide a catalog of genetic leads that may affect the traits differently in each sex for the research community who has specialty area expertise to further explore. The drug response/pharmacogenomic traits, are known to have bigger genetic effect size (10.1126/scitranslmed.3003471) when compared to human disease complex traits and is within our group’s research expertise. We believe the drug response phenotypes are more suited for laboratory testing. That is one reason we chose to study the DMET genes, which have known role in processing both endogenous and exogenous substrates. Indeed, through web-scraping, we identified thousands of drugs that may be affected by 20 differentially expressed DMET genes. When focusing on one of these gene, *CYP1A2*, closed to 50 drugs which were known to be metabolized by this enzyme and have had reported sex differences in drug response. By employing experimental procedures like western blotting, human liver microsomal incubation and LC/MS quantification, we have established the proof-of-concept that different expression of *CYP1A2* gene in livers from men and women donors are true. We also confirmed the different *CYP1A2* activities between the two sexes towards metabolizing clozapine, a widely used anti-

psychiatric drug. Beyond our own experiments, we have now added additional literature evidence that support findings from our study.

For example, in Results “Our analyses identified a number of sex-different drug responses that were supported by existing literature. For example, flunarizine, another CYP1A2 substrate, which is used in treating epilepsy (10.1248/bpb.19.1511), was reported to not affect catalepsy in male mice, but attenuated catalepsy in females at the same doses (10.1016/S0278-5846(98)00102-X). Another study found that male rats formed two oxidative metabolites of flunarizine at a higher rate than female rats (PMID: 1462051). These are in agreement with our findings that higher CYP1A2 expression in males can lead to faster metabolism/breakdown of this drug and therefore less response. Similarly, female mice have been reported to be less susceptible to acetaminophen overdose induced hepatotoxicity than male mice (10.1016/j.tox.2011.05.018). Acetaminophen is also metabolized by CYP1A2 to N-acetyl-p-benzoquinone imine (NAPQI), known to induce hepatotoxicity. In this case, the lower expression of CYP1A2 in female livers would lead to lower production of the toxic metabolite and therefore partially explain the lower toxic response in females.”

Comments on presentation of the data:

The text in many of the figure panels is extremely small and not accessible to the reader. Particularly frustrating is the illegibility of labels for the groups along both the x and y axes of key data summary figures such as Fig. 1E, 2C, and 3D. These are nicely conceived schemes to represent complex data, but are illegible without magnifying to 200%.

Response: Thank you for the comments to improve the readability of our work. We have re-created the Fig 1C, Fig 1E and Fig 2C by increasing the font size for all labels, and added higher resolution figures for the revised manuscript.

The text could benefit from editing to ensure that meaning is conveyed as clearly as possible. For example, the sentences at lines 66–70 are not stated well (do the authors mean “...DMET genes are not limited to mediating the blood concentration of xenobiotics, but also determine the amount of....a comprehensive study of sex differences in DMET gene activity and their health impact...”?). Other instances occur throughout the text.

Response: Thanks for the suggestion. We have edited the manuscript accordingly.

Reviewer #2 (Remarks to the Author):

Really important lens of looking at sex differences through DMET genes – and I think this is an excellent way to approach examining sex-specific genetic effects and is an exciting lens. I appreciated the thoroughness of genetic analysis methods used (heritability, GWAS, eQTL, expression, MR) and the follow up in liver microsomes. However, I think the lens could benefit from additional justification (by comparison with non-DMET regions), and the sex differences

results require further comparison to the literature and acknowledgement of potential contributing covariates.

Response: We appreciate the enthusiasm this reviewer has toward our work and the constructive recommendation made to further improve our work. Please see below for the details on the additional analyses performed and findings from them per reviewer's request.

Justification of the DMET lens and comparison to overall results

Response: The detailed method on traits and gene region selection can be found in Materials and Methods section under "data collection". To further clarify the exact methods employed in our study, we have modified this section along with other areas throughout the revised manuscript.

Regarding the rationale to focus on DMET genes, as in our response to reviewer 1's comment above "...when dealing with a wide-open topic with thousands of traits, tens of thousands of genes for potential evaluation, we chose to focus on DMET genes to comprehensively evaluate their sex-specific genetic regulation and health impact. The discovery pipeline employed in our study can serve as a roadmap to evaluate any other genes/pathways. Further, the smaller list of genes in our study will allow sufficient statistical power for our sex stratified analysis."

Additional reasons to focus on DMET genes were due to their functional importance in breaking down both endogenous and exogenous substrates, all of which have a firm role in human health (these are stated in the Introduction section). Note, findings from our work should be interpreted in the context of chosen gene sets and phenotypes. We have carefully examined our manuscript to make sure that all interpretation of results were in the context of DMET genes and the qualitative and quantitative findings around this set of genes were not extended or generalized to other non-DMET genes.

With these said, we agree with the reviewer that the non-DMET gene regions, although not the focus of this work, are equally important. Therefore, we have performed additional analyses to gain a sense of the non-DMET genetic regulation on traits showing sex differential genetic regulations as described below.

Particularly, in results section 1 – you examine heritability of traits with at least one significant DMET region SNP.

(1) How many traits did you start with?

Response: In the original manuscript, we started with 564 traits that have sufficient case numbers in both sexes in UKBB. These traits also have to have DMET region relevancy by having at least one SNP in the region that have been reported to be associated with them in the GWAS catalog. This list of traits was then narrowed down by estimating genome-wide heritability (h^2) separately in male and female. Only those 83 traits that show global h^2 differences between the two sexes were further evaluated. Note that sex different global h^2 estimation were derived from both DMET and non-DMET regions.

(2) How was the set of DMET genes selected? What is defined as a DMET region?

Response: see Materials and Methods section under "data collection". Briefly, we obtained the list of DMET genes from a previous publication (10.1371/journal.pone.0060368). As stated in our response to reviewer 1, we have added the detailed description of these DMET genes into a new supplementary table (Table S1).

We defined the genomic region for each DMET gene as the region 1Mb up/down stream of the transcription start site (TSS) of each DMET gene of interest.

(3) What is the reasoning behind filtering traits in this way? Would it not make more sense to estimate sex-specific DMET region heritability vs overall heritability of traits that have a DMET SNP?

(4) How does the metric for sex differences in heritability among these traits (14.7%) compare to that for 564 randomly selected traits (that don't have a DMET region SNP)? This comparison

(5) Could you partition the heritability of traits into DMET region vs non-DMET region? I think this could yield additional insights with background would justify the lens you're using.

Response to comments (3)-(5): We appreciate reviewer's thoughtful suggestion on how to potentially highlight/justify the importance of the DMET regions and have attempted to do so through the following steps. First we expanded our analysis to additional 1222 non-DMET region related human complex traits to assess/quantify global genomic impact for each trait within each sex. This step would allow us to assess whether focusing on DMET related traits would enrich for sex differentially regulated phenotypes (overall, we did not observe this, see below for details). Subsequently, we employed LAVA, one of the latest tools published in Nature Genetic (10.1038/s41588-022-01017-y), to estimate regional heritability for both DMET and non-DMET regions. We did this regional h^2 estimation for four traits: two (Gout and Hypothyroidism) show sex differences in global h^2 between sexes and were DMET related; and the other two (Acquired deformities and Inguinal hernia) also show sex differences in global h^2 between sexes and were not DMET related. Once again, we observed that the pattern of relationships between genotype/phenotype in different regions and in each sex is highly trait dependent. Therefore we made sure all findings in our manuscript were interpreted in their rightful context.

Specifically, regarding traits selection, as responded above, the 564 traits chosen in the original manuscript were those that have sufficient cases number in both sexes in UKBB and have been reported to relate to DMET regions. Among these traits, 14.7% show sex differences in genome-wide heritability. In this revision, we performed genome-wide h^2 estimation on additional 1222 traits that have sufficient samples in both sexes in UKBB yet do not have SNP-associations in DMET regions (non-DMET related traits). For these traits, we observed 13.7% (167/1222, Figure below) of them showing different h^2 between males and females, which is comparable to results obtained from DMET related traits.

Furthermore, these quantitative observations on percentage of traits that are showing sex differences in global genomic regulations are also comparable to an independent study published recently in Nature Genetic (10.1038/s41588-021-00912-0), where 71/530 (13.40%) traits showed significant differences in their heritability between two sexes.

Taken together, our original trait selection criteria which focused on a collection of DMET region related traits identify similar ~10%-15% of complex traits that have a different global genetic impact between the two sexes when compared to traits that are not known to be related to DMET regions in UKBB. We added these to the Results section as "When expanding the genome-wide heritability analysis to additional 1222 traits that have sufficient samples for both sexes in UKBB and are not known to related to DMET genetic regions, we found 13.7% (167/1222) of these traits showing sex differences in global heritability (Fig. S2, Table S4). Interestingly, similar 13.40% (71/530) traits showing significant differences in their heritability between two sexes have been reported by an independent study."

Per reviewer's request, we employed LAVA, one of the latest tools published in Nature Genetic (10.1038/s41588-022-01017-y), to assess regional heritability. We estimated regional h^2 for both DMET gene regions and non-DMET gene regions. For the latter, we randomly selected similar length non-DMET gene regions as DMET gene regions for 1000 times to generate a potential regional non-DMET regions h^2 distribution. The regional h^2 was calculated by summing the h^2 for all genes in either DMET or non-DMET gene regions. A total of four traits that are showing sex different global genomic regulations were evaluated. They are gout and hypothyroidism, which were related to DMET; and acquired deformities of fingers and toes and inguinal hernia, which were not related to DMET.

For gout and hypothyroidism (see Figures below), differential regional DMET h^2 contributions to the traits were observed between the two sexes. For gout, the DMET regions h^2 falls within the range of randomly selected non-DMET region h^2 distributions in males; while DMET regional h^2 is much lower than non-DMET regions h^2 in females. For hypothyroidism, we observed h^2 from DMET regions is greater than non-DMET region in females, but lower than non-DMET regions in males (Figure below, the dash line representing the sum of h^2 from DMET region, the histogram representing the distribution of randomly selected non-DMET

regions). There regional h2 findings are in concordance with our sex-differential effects (SDEs) analysis results reported in the main text.

For the other 2 traits (Acquired deformities and Inguinal hernia) that do not have DMET relevancy, we observed that h2 from DMET regions is lower than non-DMET regions in both sexes. These observations suggest that for these other traits, although global genetic regulation may be different between males and females, the genetic drivers for such differences are unlikely to come from the DMET regions. This further supports our initial workflow pipeline by selecting DMET related traits first to enable discoveries of genetic factors that contribute to the traits observed sex differences.

The same is true for the DMET lens in general -- e.g. line 169 you mention you characterized sex differences at a genome-wide and in DMET regions, but I see no mention of the genome-wide examination or comparison with the DMET region results. I also want to see this justification for the MR analysis.

Response: Regarding the sex-stratified MR test between biomarker and outcomes, we performed our MR test in 2 steps. First, we perform MR test using all SNPs. This is because the nature of the MR test is to identify traits-traits relationship. Only considering the DMET region could result in false discovery. Results from step1 have been reported in Figure 3C& Figure 3D, Table S11. Second, we selected sex-specific MR relationships from step 1 and performed MR test by only including SNPs from DMET regions. The rationale is to further discover potential causal relationships that are driven by genes in DMET regions, under the condition that such MR relationships exist at genome-wide scale. We reported DMET specific MR relationships in Figure S10 and Table S12. Furthermore, we provided one example of female specific MR relationship that might in part be driven by one DMET gene.

Our goal in this work is to show that variants within DMET regions can be causal for human complex health traits. Even though the MR test on genome-wide scale could conclude genetic causal signals come from both DMET and non-DMET regions, without comprehensively studying the non-DMET regions, we are not confident to draw conclusions for the non-DMET regions, especially because these conclusions are highly traits dependent.

eQTL analysis -- how many eQTLs did you examine for sex-diff and sex-spec effects

Response: The eQTL analysis were performed using GTEx data, which contains 1,419,634 SNP-gene associations shared in both sexes. For sex-differential cis-eQTLs, we start with 120 eQTLs that are significant after multiple testing correction in either or both sexes, we then performed z-score test on these eQTL to test the differential effect between sexes. For sex-specific cis-eQTL, we defined eQTL that only significantly in one sex but not the other and not fall into the sex-differential eQTL groups as defined above.

More acknowledgement of results to date in liver GEX studies

The literature search does not add much to the paper, instead a comparison to previously reported sex differences in liver expression would improve the results.

- multiple papers on sex-diff DMET genes (e.g. Yang L, Li Y, Hong H, Chang CW, Guo LW, Lyn-Cook B, Shi L, Ning B. Sex Differences in the Expression of Drug-Metabolizing and Transporter Genes in Human Liver. *J Drug Metab Toxicol*. 2012; Zhang Y, Klein K, Sugathan A, Nassery N, Dombkowski A, Zanger UM, Waxman DJ. Transcriptional profiling of human liver identifies sex-biased genes associated with polygenic dyslipidemia and coronary artery disease. *PLoS One*. 2011) -- how do your results compare?

Response: We appreciate the suggestion. In our original study, we have validated our finding through an independent dataset with 14 out of 19 DE genes (74%) showing sex differences in that independent dataset. When compared to the two publications suggested by the reviewer, we observed 8 out of 20 (*J Drug Metab Toxicol*) and 5 out of 20 (*Plos One*) of our DE genes that have been identified showing sex differential gene expression profile in these 2 publications.

Given the independent nature of these studies and a number of potential confounders (e.g., sample collection procedure, donor race, age, etc) exist among studies, this is to be expected. For example, in our analysis, we controlled for known technical confounders, such as RIN score, as well as hidden confounders which identified by SVA methods. When compared between these two listed studies, only 18% (14/77) DE genes from the first publication are verified in the second one. In the revision, we provide an additional supplementary table (Table S16) to summarize verification performance in all three independent studies. This was stated in the manuscript as “When assessing the reproducibility of our discovery in an independent dataset (ref), we recapitulated the differential expression for 14 of 19 genes (expression of *CYP1A2* was not quantified in the validation dataset). Two additional smaller datasets (refs) were also evaluated and our top differential expressed genes, such as *UGT2B17*, *UGT2A3*, *CYP3A4*, *SLC3A1*, *SLC16A14* are concordance with previous finding (Table S16).”

- the CYPs examined (1A2 and 3A4) in HLMs have known sex differences in expression in liver (you cite Waxman and Holloway earlier, but this should be acknowledged when you do the examination

Response: Acknowledgment added as “We confirmed higher protein abundance of CYP1A2 in the male pooled HLMs and the opposite trend for CYP3A4 (Fig. 4B), which is correlated with the previous report. (10.1124/mol.109.056705).”

Acknowledgement of non-genetic sex-gender differences that may contribute to results

- Many of the SDE traits that come up (hypothyroidism, gout) in sex differential heritability are (1) self-reported (there are known gender differences in reporting behaviors) and (2) have known sex differences in incidence (hypothyroidism is more common in women, gout in men). How do you account for this?

Response: (1) We shared the reviewer’s concern towards self-reported traits that, in comparison to medically diagnosed diseases, self-reported traits may introduce subjective bias. Yet, when we closely examine the data, we observed high concordance of GWAS results from self-reported and EHR identified traits (two examples of hypertension, hypothyroidism figures are shown below, data from IEU open GWAS project, <https://gwas.mrcieu.ac.uk/>) using the UK biobank data. Furthermore, the field has argued that both ICD code identified traits and self-reported traits have their own values. For example, mild symptoms may not be recorded/diagnosed in the EHR. Knowing these, we have made sure that we clearly reported our results to reflect the sources of trait identification. We also added a note in the Discussion section to caution readers on interpreting results based on how the phenotypic traits were collected.

(2) The reviewer is correct regarding there is imbalance in the sample size between the two sexes when sex difference in incidences was known. The differences in different case numbers for those case/control traits could affect the power of GWAS discovery. We were aware of this issue at the beginning of our analysis. Therefore, we only selected traits that have cases number > 300 in both sexes for subsequent analysis to avoid simply missing findings due to small sample size in one sex. We acknowledge that this might not be enough to avoid false positive discovery, we argue that such sex dimorphism in disease incidences highlights the differences in disease etiology, which could be reflected in GWAS results. Previously, sex differences in autosomal allele frequency were only reported in 12 genes (10.1007/s13258-015-0332-z), where DMET genes are not among them. We also compared our results with sex-stratified GWAS in 11 traits which were analyzed using REGENIE (logistic Mix-Effect Model) which accounts for case-control imbalance. We observed a great concordance from both results in h^2 , genetic correlation, and the number of SNPs with SDEs (Discussion).

In an ideal scenario, GWAS could be performed after matching the sample size between males and females. However, having access to only the summary statistics from the UKBB prevented us from taking this approach.

Together with comment (1), we have added limitations of our study in the Discussion "Sex differences in genetic effects could be confounded by both sociological and behavioral differences between males and females. Such differences complicate the detection of true molecular mechanisms of disease. We noted that the potential sex differences highlighted in our results need to be distinguished with many features of behavior and external environments in future investigations."

- in discussion of alcohol intake locus (1221-234) -- there are known sex-gender differences in alcohol consumption that are not necessarily genetic, and relate to body size and sociocultural patterns, make sure to acknowledge this

- Same is true re incidence and behaviors for many of the traits mentioned in figure 2

Response: Thanks for the suggestions. We agree with the reviewer, and added to the results “Of note, the observed sex dimorphism in alcohol consumption can be affected by factors, such as body size, sociocultural behavior. The identification of genetic contributors to this trait should not be interpreted without these other factors. Our results provided a plausible explanation for this phenomenon where sex-differentiated genetic regulation of *ADH1C* may play a part in this complexed issue.”

MR analysis needs more examination

For the MR analysis, it is not clear which SNPs are used as instruments. It is important that the instruments are not selected from the same GWAS as the MR analysis is performed on -- and it is not clear if this is the case. Also there are a couple cases where the results could be affected by winner's curse: first, "traits were selected based on whether there are shared significant variants between exposure and outcome that are mapped into the DMET genes region", and second, a follow up MR analysis was performed only looking at "SNPs in DMET regions in traits with sex-specific causal effects". Follow up analysis in a validation cohort is required.

Response: To clarify our analytical pipeline, we first performed MR analysis between serum biomarker traits (exposures) and complex traits (outcomes) using MR Egger regression considering genome-wide SNPs. For each MR test, the instrumental variables (SNPs) are selected from the exposure traits. Only SNPs have significant SNP-trait association ($p < 5 \times 10^{-8}$) were selected as instrumental variables. Results from this step have been reported in Figure 3C& Figure 3D, Table S11. Subsequently, we selected sex-specific MR relationships from the first step and performed MR test by only including SNPs from DMET regions. The rationale is to further discover potential causal relationships that are driven by genes in DMET regions, under the condition that such MR relationships exist at genome-wide scale. We reported DMET specific MR relationships in Figure S10 and Table S12.

Although we mainly presented sex-specific potential causality in our manuscript, we also identify a number of, clinical validated, causal relationships in both men and women. For example, we found causal relationship in Urate-self-report: Gout ($p = 3.15e-61$); Glycated hemoglobin-self-report: diabetes ($p = 3.78e-46$); Gamma glutamyl transferase- High cholesterol ($p = 5.86e-39$); HDL-High blood pressure ($p = 2.53e-24$), etc. These results in-part support the validity of our MR analysis. Second, we applied our MR analysis using MR Egger regression, which can provide a causal effect estimate which is not subject to the violation of Independence assumption (10.1093/ije/dyv080).

We provide an example that testosterone increases the risk of high blood pressure in females but not in males. This finding is supported by multiple literature evidences (10.1097/HJH.0b013e3283603eb1, 10.2147/CIA.S195498, 10.1152/ajpheart.00681.2014, 10.1161/HYPERTENSIONAHA.111.180620). Through colocalization, we observed that testosterone and high blood pressure are likely to share the same causal locus in females only. This locus mapped to *CYP11B1* gene which previously has been associated with hypertension (10.1073/pnas.90.10.4552). Our analysis highlights the utility of MR and colocalization in identifying potential causality.

In additional to what we reported in the manuscript, we also provide a number of MR results that showing sex differences. Some of them have literature support. For example, we observed a male-specific MR results between Apolipoprotein B (ApoB) and coronary heart disease.

Although ApoB increase the risk of coronary heart disease has been reported in both male and female, a sex-differential effect has been observed. Specifically, with the same ApoB baseline level, female takes longer time than male to experiences heart disease (Age45-54, F:18 years, M:10 years; Age 65-74, F:5 years, M:2 years, 10.1371/journal.pmed.1003853). This highlights the potential differences of ApoB in increasing the risk of heart disease between sexes. LocusZoom demonstrated that ApoB level and coronary heart disease share the same LPA gene locus in males but not in females. This indicates a potential different mechanism of ApoB in heart health between sexes or protecting effects that are unique in females.

MR test is heavily affected by the confounding factors, whereas sex differences in MR analysis could be more confounded by environmental factors that is different between male and female. To identify potential validation datasets, we have searched literature and as much biobank data as possible. However, our experience showed that it is very rare to find sex stratified GWAS summary statistic. In the absence of independent validation dataset, we made sure that we do not claim direct causality, but position our findings as a way to point potential directions to study sex-differentiate/specific molecular mechanism that has been masked in sex-combined study.

Minor notes:

The paper would benefit with clearer justification of bridge between DMET genes and other traits of interest (e.g. in the abstract -- mention high BP without making the jump to why we are looking at non-drug traits, remove and make this clearer in the abstract or intro)

Response: We have modified the Abstract and the Intro accordingly.

Clearer distinction between sex-differential and sex-specific effects - clear the authors understand, but define this earlier (e.g. introduction). Also for places where you just mention the sex-specific effects (e.g. causal loci line 147), also test for differences.

Response: Thanks for the suggestions. We have added in the introduction “We tested both sex-differential genetic effect where the effect-size of genetic regulation is different between sexes, and sex-specific genetic effect where the association is only significant in one sex.”

For line 147: this sex-specific genetic effect is defined based on the colocalization results. The sex-specific genetic effects are at the same loci, as with the trait showing Posterior probabilities to be causal only in one sex but not the other. Therefore, there is no effect-size to be tested for differential effect. We have applied the sex-differential effect carefully in the eQTL analysis.

- title: I would change "decipher" to "deciphering"

Response: Correction made.

- abstract could be clearer, I found it hard to follow (e.g. remove "For example" line 20)

Response: Requested modification made.

- introduction lines 63-66 -- should acknowledge that PGx studies often do not consider sex because of size/power limitations (and generally acknowledge this as a problem for sex-separated or sex-aware GWAS)

Response: Thanks for the suggestion. We have added in the introduction “However, these studies were typically conducted in both sexes and report results in a sex-combined fashion, because of the sample size/power limitations, hence underestimating the role of sex as a modifier of the drug response.”

- line 83 -- multiple previous papers on this! cite them

Response: We have now added reference to support the statement.

- line 90 -- sex differences in genetic architecture *of DMET genes* <-- this is what you were looking at

Response: As we also conduct analysis considering genome-wide h^2 and genetic correlation, we modified the subtitle to “Sex differences in global genetic architecture and in DMET gene region.”

- line 137 "However" sentence is not a complete sentence

Response: We have made changes accordingly. “However, whether these differential genetic effects are functional related to human health is unknown.”

- Figure 1 -- could not read the figure, resolution is too low

in part D: "traits show" not shows

Response: We have re-created Fig 1C, Fig 1E and Fig 2C, increased the font size of labeling and added higher resolution figures for the revised manuscript.

- what are sex heterogeneity SNPs? define this

Response: We re-created the figure now changed to SNP with SDEs.

- line 223 "Therefore" - this does not directly follow and requires a citation

Response: We have made changes accordingly. 1007/s00439-012-1163-5)

Supplemental Table legends - please describe the columns in more detail in the "Meta" sheet or elsewhere.

Response: Descriptions for all supplementary tables have now all been improved, with column descriptions added.

- line 253 citation fo sex differences in serum biomarkers is about testosterone specifically, include other citations that describe these differences

Response: We have additional reference to support our statement (10.1161/CIRCULATIONAHA.116.023005).

- line 372: PharmGKB and DrugBank require citations

Response: We have added citations for PharmGKB (10.1002/cpt.2350) and DrugBank (10.1093/nar/gkj067).

- I think you can add more lead up to the micorosome analysis -- this is a strength of the paper but was hard to follow when mentioned on line 365-368. Mention that you did this as a follow up analysis. May want to move to another section

Response: We have modified the section “We further quantified the protein abundance of two pharmacogenes, CYP1A2 and CYP3A4, using pooled human liver microsomes (HLMs). The HLMs were collected separately from male and female donors and the protein abundance was quantified using western blot. We confirmed the higher protein abundance of CYP1A2 in the male-pooled HLMs and the opposite trend for CYP3A4 (Fig. 4B), which is correlated with the previous report.”

- line 383: "high intensity" is not the correct word

Response: Change made.

- line 480-481: I am not sure what you are referring to here?

Response: We have clarified the sentences to “Interestingly, on average, 40% of these causal relationships remain significant when only including the DMET SNPs as instrumental variables.”

REVIEWERS' COMMENTS

Reviewer #1 (Remarks to the Author):

The authors have provided a response to all comments; however, in some cases this involves stating that further functional validations are beyond the capability of their group. Changes to the manuscript include a table of the DMET genes analyzed (which both reviewers requested), inclusions of comparisons of some non-DMET genomic regions with the DMET regions, and inclusion of additional references that support a role for CYP1A2 in sex-biased drug responses. The authors have also made some changes in presentation of the data, as suggested in the review.

Reviewer #2 (Remarks to the Author):

The response to reviewers is thorough and addresses many of the concerns.

The authors provide additional justification for use of DMET genes, as well include their definition and list of genes in a supplemental table, which is important for the paper. Figures and tables, as well as their legends are improved and now much easier to examine.

They also provide a thorough response to reviewers' suggestions to compare DMET findings to non-DMET genes, examining heritability, genetic correlation, and in MR analysis. It was interesting to see that focus on DMET did not enrich for sex-differential phenotypes, and this is an important addition. I also appreciate the comparison with the literature fraction of differences.

The authors added an important note about possible confounding due to sociocultural factors, as well as justify the use of self-reported summary statistics. Please also mention sex differences in incidence of these traits (hypothyroidism, gout) in the article text, and the fact that they could not be matched because of use of summary statistics. This is mentioned in the response to reviewers but not included.

minor (wording/refs):

- abstract: "a subset of DMET genes" -- change this to two
- the 13.4% statistic in an independent study needs a reference
- "Two additional smaller datasets (add refs) " <-- make sure to add these refs

in most cases, change "genetic effect"  "genetic effects"

"DMET gene region"  "DMET gene regions"

"are functional related"  "are functionally related"

"play a part in this complexed issue"  "play a part in this complex issue"

"are concordance with previous finding"  "are concordant with previous findings"

"a number of sex-different drug responses"  "sex-differential"

"correlated with the previous report"  "correlated with a previous report"

Responses to the Reviewers' Comments

NCOMMS-22-24647A

Title: Deciphering genetic underlying causes for sex differences in human health through the lens of drug metabolism and transporter genes

REVIEWERS' COMMENTS

Reviewer #1 (Remarks to the Author):

The authors have provided a response to all comments; however, in some cases this involves stating that further functional validations are beyond the capability of their group. Changes to the manuscript include a table of the DMET genes analyzed (which both reviewers requested), inclusions of comparisons of some non-DMET genomic regions with the DMET regions, and inclusion of additional references that support a role for CYP1A2 in sex-biased drug responses. The authors have also made some changes in presentation of the data, as suggested in the review.

Response: We thank the reviewer for taking the time to go through our manuscript once again, and we are pleased that the large majority of the reviewer's concerns are now sufficiently addressed.

Reviewer #2 (Remarks to the Author):

The response to reviewers is thorough and addresses many of the concerns.

The authors provide additional justification for use of DMET genes, as well include their definition and list of genes in a supplemental table, which is important for the paper. Figures and tables, as well as their legends are improved and now much easier to examine.

They also provide a thorough response to reviewers' suggestions to compare DMET findings to non-DMET genes, examining heritability, genetic correlation, and in MR analysis. It was interesting to see that focus on DMET did not enrich for sex-differential phenotypes, and this is an important addition. I also appreciate the comparison with the literature fraction of differences.

Response: We thank the reviewer for recognizing our efforts to address the reviewer's concerns.

The authors added an important note about possible confounding due to sociocultural factors, as well as justify the use of self-reported summary statistics. Please also mention sex differences in incidence of these traits (hypothyroidism, gout) in the article text, and the fact that they could not be matched because of use of summary statistics. This is mentioned in the response to reviewers but not included.

Response: Per reviewer's request, we have added the following sentence to the Discussion.

“Third, for traits that exhibit sex differences in incidences (e.g., hypothyroidism), there is often imbalance in the sample size between the two sexes, which could affect the power of sex stratified GWAS discovery. This is partly why we only selected traits that have cases number > 300 in both sexes for our analysis to avoid simply missing findings due to small sample size in

one sex. We acknowledge that this might not be enough. In an ideal scenario, GWAS could be performed after matching the sample size between males and females. However, having access to only the summary statistics from the UKBB prevented us from taking this approach.”

We once again thank the reviewer for his/her time and for substantially improving the quality of this manuscript.

minor (wording/refs):

- abstract: "a subset of DMET genes" -- change this to two

Response: The validation of differentially expressed genes were done through two steps. One, independent dataset confirmation, for which we recapitulated 14 of 19 genes initially discovered in GTEx liver dataset; and two, experimental validation, for which we focused on CYP1A2 and CYP3A4, not only for their expression level but also enzyme activity. “a subset of DMET genes” in the abstract, is referenced to the replication of our initial findings in an independent dataset (Step 1). We have modified the sentence as “Furthermore, we identified and validated sex differential gene expression of a subset of DMET genes in human liver samples.”

- the 13.4% statistic in an independent study needs a reference

- "Two additional smaller datasets (add refs) " <-- make sure to add these refs

in most cases, change "genetic effect"  "genetic effects"

"DMET gene region"  "DMET gene regions"

"are functional related"  "are functionally related"

"play a part in this complexed issue"  "play a part in this complex issue"

"are concordance with previous finding"  "are concordant with previous findings"

"a number of sex-different drug responses"  "sex-differential"

"correlated with the previous report"  "correlated with a previous report"

Response: We have made all requested wording/gramma and references changes.